# Sulfur isotopes as a proxy for human diet and mobility from the preclassic through colonial periods in the Eastern Maya lowlands

Claire E. Ebert[1]*, Asta J. Rand[2], Kirsten Green-Mink[3], Julie A. Hoggarth[4], Carolyn Freiwald[5], Jaime J. Awe[6,7], Willa R. Trask[8], Jason Yaeger[9], M. Kathryn Brown[9], Christophe Helmke[10], Rafael A. Guerra[7,11], Marie Danforth[12], Douglas J. Kennett[13]

1 Department of Anthropology, University of Pittsburgh, Pittsburgh, Pennsylvania, United States of America, 2 Department of Archaeology, Memorial University of Newfoundland, St. John's, Newfoundland, Canada, 3 Department of Anthropology, Idaho State University, Pocatello, Idaho, United States of America, 4 Department of Anthropology and Institute of Archaeology, Baylor University, Waco, Texas, United States of America, 5 Department of Anthropology and Sociology, University of Mississippi, Mississippi, United States of America, 6 Department of Anthropology, Northern Arizona University, Flagstaff, Arizona, United States of America, 7 Institute of Archaeology, National Institute of Culture and History, Belmopan, Belize, 8 Defense POW/MIA Accounting Agency, 9 Department of Anthropology, The University of Texas at San Antonio, San Antonio, Texas, United States of America, 10 Institute of Cross-Cultural and Regional Studies, University of Copenhagen, Copenhagen, Denmark, 11 Department of Anthropology, University of New Mexico, Albuquerque, New Mexico, United State of America, 12 School of Social Science and Global Studies, University of Southern Mississippi, Hattiesburg, Mississippi, United States of America, 13 Department of Anthropology, University of California, Santa Barbara, Santa Barbara, California, United State of America

* c.ebert@pitt.edu

**Data Availability Statement:** All relevant data are within the manuscript and its S1 and S2 Tables and S1–S3 Texts files.

## Abstract

Maya archaeologists have long been interested in understanding ancient diets because they provide information about broad-scale economic and societal transformations. Though paleodietary studies have primarily relied on stable carbon ($\delta^{13}C$) and nitrogen ($\delta^{15}N$) isotopic analyses of human bone collagen to document the types of food people consumed, stable sulfur ($\delta^{34}S$) isotope analysis can potentially provide valuable data to identify terrestrial, freshwater, or marine/coastal food sources, as well as determine human mobility and migration patterns. Here we assess applications of $\delta^{34}S$ for investigating Maya diet and migration through stable isotope analyses of human bone collagen ($\delta^{13}C$, $\delta^{15}N$, and $\delta^{34}S$) from 114 individuals from 12 sites in the Eastern Maya lowlands, temporally spanning from the Late Preclassic (300 BCE—300 CE) through Colonial periods (1520–1800 CE). Results document a diet dominated by maize and other terrestrial resources, consistent with expectations for this inland region. Because $\delta^{34}S$ values reflect local geology, our analyses also identified recent migrants to the Eastern lowlands who had non-local $\delta^{34}S$ signatures. When combined with other indicators of mobility (e.g., strontium isotopes), sulfur isotopic data provide a powerful tool to investigate movement across a person's lifespan. This study represents the largest examination of archaeological human $\delta^{34}S$ isotope values for the Maya lowlands and provides a foundation for novel insights into both subsistence practices and migration.

**Funding:** Funding for previously unreported carbon, nitrogen, and sulfur isotopic analyses presented in this paper was provided by Baylor University (JAH), The Pennsylvania State University (DJK), the University of Pittsburgh Center for Comparative Archaeology (CEE; https://www.comparch.pitt.edu/), the Canadian Association for Physical Anthropology (AJR; https://capa-acap.net/), and Social Sciences Research Council of Canada (#767-2014-2712; AJR; https://www.sshrc-crsh.gc.ca/home-accueil-eng.aspx). The funders had no role in study design, data collection and analysis, decision to publish, or preparation of the manuscript.

**Competing interests:** The authors have declared that no competing interests exist.

# Introduction

Paleodietary studies are critical for archaeologists examining past economic and social transformations since what people ate reveals how they interacted with both the natural and social world. While an array of archaeological techniques exists to gather information about the food resources available to past human societies (e.g., paleoethnobotany, zooarchaeology) [1, 2], stable isotope analyses of human skeletal tissues directly reflect the types of foods people consumed [3]. Isotopic data correspond to the local ecology and allow for the reconstruction of broad dietary shifts across populations, including the adoption of agriculture and adaptations to climate change [e.g., 4–9]. The data are specific to an individual, providing a means to explore the social dimensions of food acquisition and consumption choices among prehistoric communities [e.g., 10–15].

Stable isotop paleodietary studies from the Maya lowlands have analyzed over 1000 human burials, providing compelling evidence for dietary change and regional variability in the region over the last 10,000 years [8, 15–17]. This research has primarily been informed by carbon ($\delta^{13}C$) and nitrogen ($\delta^{15}N$) analyses of bone collagen, which record the sources of plant and animal protein that people consumed, and $\delta^{13}C$ values from bone apatite, which record the mix of protein and carbohydrates in the diet [18].

Despite being one of the most comprehensively studied regions of the ancient world for isotopic studies, however, applications of sulfur isotope ($\delta^{34}S$) analyses to the reconstruction of diet and mobility patterns have not been well investigated in the Maya lowlands. This is due in part to relatively poor preservation of human skeletal tissues at many lowland sites [19], as well as the necessity of a comparatively large sample of well-preserved bone collagen for $\delta^{34}S$ analyses [20]. Sulfur isotope values, however, can be valuable in paleodietary studies because they distinguish between food resources from terrestrial, freshwater, and marine ecosystems, complementing $\delta^{13}C$ and $\delta^{15}N$ collagen and $\delta^{13}C$ carbonate data [21–24]. Because $\delta^{34}S$ values in bone collagen correlate with local geology, these data can also be used to track migration during the last years an individual's life, supplementing information about population movement from archaeological or other isotopic evidence [25–30].

We report the results of stable carbon, nitrogen, and sulfur isotope analyses from 114 human individuals from the Eastern Maya lowlands, focusing on sites in the upper Belize Valley of western Belize and adjacent areas, to evaluate the utility of sulfur isotope data in ancient Maya paleodietary studies (Fig 1). This study, which expands upon previously published data by contributing new isotopic values from additional samples, represents the largest examination of archaeological human sulfur isotopes in the Maya lowlands to date. It is also the first to comprehensively analyze the links between $\delta^{34}S$ values, diet, and human movement within a single region of the lowlands. The samples come from 12 sites spanning the Late Preclassic (300 BCE—300 CE) through Colonial periods (1520–1800 CE; Table 1), providing a long-term perspective on diet and mobility in the Eastern Maya lowlands. While our results indicate diets were likely stable over the course of 2000 years in the Eastern lowlands, trends in $\delta^{34}S$ data suggest that some Eastern lowland communities strategically accessed food resources from different ecosystems. Additionally, when compared to previously published strontium ($^{87}Sr/^{86}Sr$) data, our $\delta^{34}S$ data identified non-local migrants in the sample, indicating the effectiveness of this method for examining patterns of migration and mobility across an individual's lifetime in the Maya lowlands. These findings demonstrate that stable sulfur isotope analysis, when used in conjunction with other isotopic assays, contributes novel insights into both subsistence practices and migration among the Maya of the Eastern lowlands and the surrounding areas.

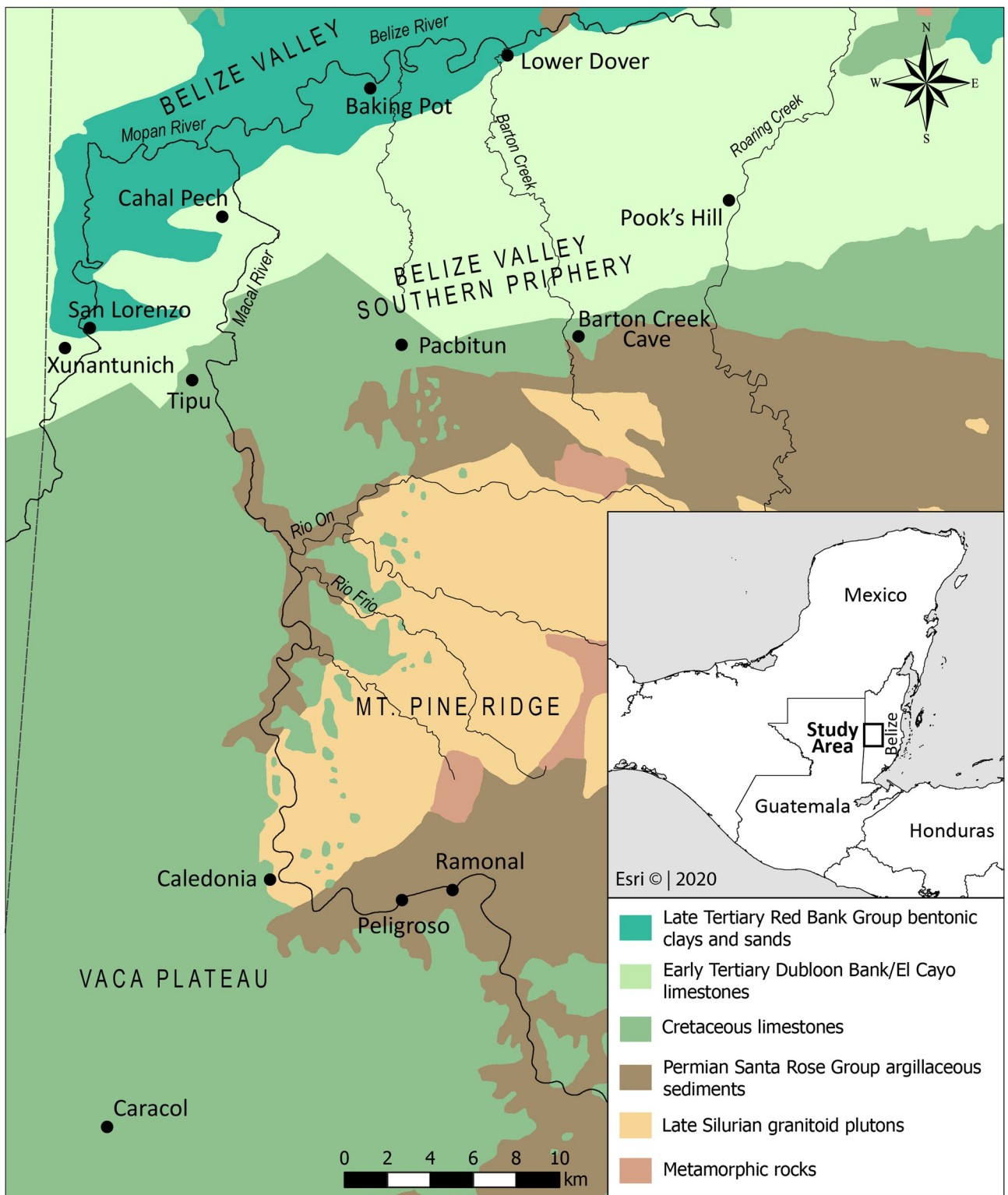

**Fig 1. Location of sites in the Eastern Maya lowlands and corresponding bedrock geology (after [95]).** Base map images are the intellectual property of Esri and are used herein under license. Copyright 2020 Esri and its licensors. All rights reserved.

**Table 1. Time periods, associated age ranges, and sample sizes for burials (total $n$ = 114) analyzed in this study.**

| Time Period | Date Range | Number of Individuals |
|---|---|---|
| Late Preclassic | 300 BCE—300 CE | 5 |
| Early Classic | 300–600 CE | 9 |
| Late Classic | 600–750/800 CE | 61 |
| Terminal Classic | 750/800–900 CE | 14 |
| Early Postclassic | 900–1250 CE | 2 |
| Late Postclassic | 1250–1520 CE | 10 |
| Colonial | 1520–1800 CE | 12 |

Note that one sample could not be assigned to a time period.

## Stable isotopes and paleodietary reconstruction in the Maya Lowlands

Stable isotope measurements of human bone collagen are a well-established proxy for prehistoric human diet because they reflect the types of foods consumed in the last 5 to 10 years of an individual's life [31]. Stable carbon ($\delta^{13}C$) and nitrogen ($\delta^{15}N$) isotope measurements are especially useful for documenting the sources of dietary protein consumed by individuals [18, 32, 33], and have been widely used by Maya archaeologists over the past 30 years to examine questions regarding the relationships between diet, the natural environment [e.g., 6, 8, 34], sociopolitical roles and status [e.g., 17, 15, 35], and economy [e.g., 13, 16, 36].

Carbon isotope values ($\delta^{13}C$) provide proxies for dietary protein based on the isotopic composition of animals and plants that an individual consumed. The $\delta^{13}C$ values of plants are determined by the photosynthetic pathways used by $C_3$ (Calvin-Benson) and $C_4$ (Hatch-Slack) plant species [37, 38]. Plant $\delta^{13}C$ values are reflected in the tissues of human and animal consumers through metabolic fractionation in which bone collagen $^{13}C$ is enriched by ~+5 ‰ relative to plants [39]. The natural vegetations of the Maya lowlands are dominated by $C_3$ plants (trees, shrubs), which possess distinctively negative $\delta^{13}C$ values averaging –26.5 ‰ [40, 41]. Local environmental conditions can introduce variation in the $\delta^{13}C$ values of $C_3$ species, however, including canopy density (i.e., "the canopy effect"; [42, 43]), resulting in $\delta^{13}C$ values for consumer tissues of approximately –22 to –18.5 ‰ [44, 45].

The most abundantly consumed $C_4$ plant in the Maya lowlands was domesticated maize (*Zea mays*), which is characterized by less negative $\delta^{13}C$ values (~–12.0 to –10.0 ‰ [46]). Domesticated maize also has $\delta^{13}C$ values +1–2 ‰ higher compared to wild $C_4$ grasses and CAM plants in Mesoamerica [47], allowing researchers to track the dietary importance of maize as a staple crop based on less negative $\delta^{13}C$ values of approximately –7 ± 1 ‰ in human bone collagen [16, 48]. Though bone collagen $\delta^{13}C$ values of marine species can span between +10 to +15 ‰ based on the type of marine environment, Caribbean species show elevated $\delta^{13}C$ values corresponding to the range of terrestrial $C_4$ plants [49; see also 50, 51].

Nitrogen isotope ($\delta^{15}N$) values in human bone vary with trophic level, increasing stepwise by approximately +3 to +5 ‰ between trophic levels in terrestrial ecosystems [39, 52]. Human diets that rely on terrestrial animal species will therefore exhibit higher $\delta^{15}N$ values compared to herbivorous diets. Marine species generally have higher bone collagen $\delta^{15}N$ values relative to terrestrial vertebrates because of an increased number of trophic levels (between four to six) in aquatic ecosystems [53], though values can be impacted by inputs from terrestrial nitrates from internal oceanic mixing [54]. In some parts of the world (e.g., Europe), these variations facilitate differentiation between terrestrial food sources and marine foods from higher trophic levels consumed by prehistoric populations [55, 56]. In the Maya lowlands, on the other hand, the $\delta^{15}N$ values of marine fauna overlap with those of both wild and terrestrial animals [49, 57,

58], resulting in human bone collagen values that can mimic those of maize-based diets. Zooarchaeological data, however, suggest relatively little consumption of marine species in the Eastern lowland study area (see discussion below).

The isotope ecologies of freshwater ecosystems in the Maya lowlands are not well investigated, though available studies suggest that freshwater species have highly variable $\delta^{13}$C and $\delta^{15}$N values, reflecting species-specific dietary preferences [e.g., 49; see also 59]. Depending on local ecologies, both $\delta^{13}$C and $\delta^{15}$N values of freshwater animals can overlap with terrestrial animals and humans [16]. For example, freshwater reptiles (turtles, crocodiles) from across the Maya lowlands exhibit more negative yet variable $\delta^{13}$C values (–21.1 ± 4.8 ‰) that overlap with C$_3$-based diets of terrestrial herbivores, though they possess elevated $\delta^{15}$N values of approximately +8.7 ± 1.9 ‰ [49]. Freshwater fish have average $\delta^{13}$C values of –25.9 ± 4.9 ‰ and average $\delta^{15}$N values +11.2 ± 1.5 ‰ ($n$ = 11) [35, 49]. Meat from freshwater *jute* snails (*Pachychilus glaphyrus*; $n$ = 2) has lower $\delta^{13}$C values (–30.3 ± 2.5 ‰) and $\delta^{15}$N values (+5.2 ± 0.4 ‰) [35]. This pattern indicates that while freshwater ecosystems in the lowlands typically exhibit more negative $\delta^{13}$C values, they are characterized by additional trophic levels compared to terrestrial ecosystems [59, 60].

Additional cultural factors that potentially impact $\delta^{15}$N values include *milpa* (i.e., slash-and-burn) agriculture practiced by the Maya. For example, burning has been shown to increase the $\delta^{15}$N values of plants for up to 5 years [61–63], which in turn could potentially impact the $\delta^{15}$N of consumers. While some archaeologists hypothesize that preferred animal species, like white-tailed deer (*Odocoileus virginianus*), were hunted in the immediate vicinity of *milpas* while grazing [e.g., 64], archaeological specimens have consistently more negative $\delta^{13}$C and lower $\delta^{15}$N values in the Eastern lowlands, reflecting a predominately C$_3$ diet [6, 49, 65, 66]. Instead, elevated $\delta^{15}$N of plants procured from *milpa*s is more likely to impact human rather than animal consumers. Values for $\delta^{15}$N are also subject to shifting environmental factors, including precipitation, aridity, and soil nitrates [67]. Distinctively high $\delta^{15}$N values in terrestrial consumers have been found to correlate with arid conditions in archaeological contexts from around the world [68], a trend noted both for animals [69, 70] and humans [8, 69].

The addition of sulfur ($\delta^{34}$S) isotope analysis to complement carbon and nitrogen dietary data has become more common in archaeological studies over the past decade [20, 29, 71]. Bioavailable $\delta^{34}$S values in the environment are influenced by precipitation, ground water, and underlying geology, allowing archaeologists to differentiate between diets with distinct $\delta^{34}$S compositions. Though atmospheric and biological processes can impact $\delta^{34}$S values, bedrock geology is the primary factor that determines the amount of bioavailable sulfur to plants and animals within terrestrial, estuarine/coastal, and freshwater ecosystems [72, 73]. Terrestrial ecosystems typically possess a range of $\delta^{34}$S values (+8 to +25 ‰), related to the composition and age of the local geology. Marine ecosystems have mean $\delta^{34}$S value of approximately +21 ‰ because of continual oceanic mixing [74]. The transport of sulfur through sea spray aerosols (i.e., the "sea spray effect") can also lead to increased $\delta^{34}$S values in coastal terrestrial systems [24, 75, 76], potentially mimicking marine values up to 30 km inland ([77, 78]. This is likely not a contributing factor to $\delta^{34}$S values in the Eastern lowlands, however, which is located approximately 80 km from the Caribbean coast. Because of this distance, estuarine resources were also not an important part of Eastern lowland diets, though it should be noted that the $\delta^{34}$S value species from these environments can potentially reflect a 'aquatic' signal when seawater inputs are present [79]. Similarly, although marine fertilizers have been found to influence the $\delta^{34}$S values of cultigens in other archaeological contexts [80], it is unlikely they were used at inland Maya sites.

Freshwater ecosystems possess variable $\delta^{34}$S values (–20 to +14 ‰) resulting from the action of anaerobic bacteria in lacustrine environments and inputs from different sources of

sulfur along their courses [81, 82]. In archaeological contexts where the consumption of marine or estuarine resources is low, such as the Eastern Maya lowlands, high $\delta^{15}$N values and low $\delta^{34}$S values may instead identify the consumption of freshwater foods. This is the case, for example, for herbivores who graze on foods from wetland environments where anoxic soils high in bacterially reduced sulfides have extremely low $\delta^{34}$S values [83, 84], but higher than expected $\delta^{15}$N values, associated with nitrification-denitrification in marsh plant species [85]. The identification of freshwater resource consumption using $\delta^{34}$S data in tropical contexts like the Eastern lowlands has not been extensively investigated [see 24], so it remains unclear how lacustrine or freshwater foods are reflected in the $\delta^{34}$S values of human bone collagen. Faunal $\delta^{34}$S data recently published by Rand and colleagues [49] suggests a range of values from approximately +2 to +16 ‰, reflecting a combination of sample location (coastal vs. inland) and the feeding habits of different species. That study also found that terrestrial fauna have elevated $\delta^{34}$S values relative to freshwater species, offering the potential to differentiate the types of protein consumed by the Maya.

Because $\delta^{34}$S values of consumers differ from those of their diets only slightly (+0.5 ± 2.4 ‰; [71, 86]), they are also useful as indicators of prehistoric mobility patterns [71, 87–89]. Across the Maya lowlands, environmental $\delta^{34}$S values vary based on underlying geology, as well as distance from coastal zones related to their depositional history. Modeled $\delta^{34}$S values for the Maya lowlands based on known geology and precipitation amounts increase from west-to-east and south-to-north, from the highlands to the Caribbean coast [90]. These gradients provide the ability to identify discrete "local" $\delta^{34}$S signatures across a broad region. A preliminary study by Rand and colleagues [49] used these data to document non-local signatures of fauna. Identification of non-local $\delta^{34}$S values for humans can similarly identify individuals who migrated into a region within the last 5 to 10 years of their lives and may possibly serve as indicators of origin at the local level (i.e., within a sub-region).

## Eastern Maya lowlands regional background

### Geology

The geochemical baseline for $\delta^{34}$S values in bone collagen is driven by soluble sulfate from atmospheric precipitation, groundwater, microbial activity, but primary from geology. While the Eastern Maya lowlands receives approximately 500 to 600 mm of precipitation annually [91], this circumscribed geographic area (~1000 km$^2$) is subject to relatively persistent atmospheric conditions, and is too far inland for a sea spray effect to impact soil sulfates [77, 78]. Although bacterially reduced sulfide will contribute low $\delta^{34}$S values to foodwebs based in anoxic aquatic environments [92], processes such as geologic weathering are most likely to influence environmental $\delta^{34}$S values in humid climates such as that of the Eastern Maya lowlands [49, 93, 94].

The Eastern lowlands are characterized by heterogeneous geology, with significant variation occurring over relatively small distances [95, 96]. This subregion is dominated by Cretaceous and Tertiary limestones, except for the older Permian igneous and metamorphic sediments of the Maya Mountains (see Fig 1). The upper Belize Valley is a linear karstic depression, measuring approximately 85 km east-to-west [97], located between limestone formations flanking the Maya Mountains to the south, including the El Cayo/Doubloon Bank carbonate formations, which extend into the northwestern Petén of the Central lowlands [98]. The Vaca Plateau is located south of the upper Belize Valley proper and consists of extensive horizontally stratified limestones [99]. These older deposits contain inclusions of dolomite, marl, and gypsum [95, 98]. To the east of the Vaca Plateau lies the Mountain Pine Ridge sub-zone of the Maya Mountains. It is the largest of three major granitoid intrusions in the Maya Mountains, and dates to

the Silurian [100]. Because of the acidity caused by these soil-forming materials, the region is unsuitable for farming, but does support deciduous seasonal forests consisting mostly of oak, pine, and palm [96]. The remainder of the Maya Mountains (~80%) is characterized by Santa Rosa Group sediments composed of conglomerates, sandstones, shales, phyllites, slates, and quartzites [101]. These sediments include some of the oldest in western Belize (~3–2.5 mya).

Several major drainages crisscross the foothills between the upper Belize Valley and the Maya Mountains, depositing sulfates from their head waters and along their courses into catchments downstream, impacting locally available sulfur in these areas. The Mopan River begins in the Chiquibul Forest and runs through marine carbonates in eastern Guatemala before joining with the Macal River to form the Belize River [102]. The Macal River, Barton Creek, Roaring Creek, and the Sibun River originate in the igneous and metamorphic formations of the Maya Mountains and drain northwards, flowing over limestone and dolomite in the southern periphery of the Belize Valley before joining with the Belize River [102].

Well-defined geology, in addition to the results of limited $\delta^{34}$S analyses of animal and human bone collagen from the Maya lowlands, allows for creation of an isoscape model that predicts $\delta^{34}$S values for isotopically distinct zones of western Belize (Table 2 and Fig 2). Average values are generally expected to negatively correlate with the age of geological sediments [49] and should increase from south to north. For example, the oldest sediments in the study region derive from the Maya Mountains and are expected to have the lowest $\delta^{34}$S values (> +8 ‰) based on geological weathering [103]. Values for the Vaca Plateau are expected to be slightly higher. The $\delta^{34}$S values from 14 human bone samples from the site of Caledonia in the Vaca Plateau (also included in this study) produced a range of $\delta^{34}$S values between +7.0 and +14.0 ‰, consistent with predicted values for this part of the Eastern Maya lowlands (though overlapping with predicted values for several regions) [90, 104].

The highest $\delta^{34}$S values are predicted for the upper Belize Valley and surrounding regions, centered on the sites of Xunantunich, San Lorenzo, and Cahal Pech. While these sites are situated on the El Cayo formation, their catchment zone extends into the Red Bank geologic zone, and therefore influences their $\delta^{34}$S values. A recent study from Nakum, located ~35 km to the east of the upper Belize Valley in the neighboring Central lowlands of Guatemala, established baseline $\delta^{34}$S values for Paleocene limestones based on the analyses of archaeological fauna [66]. The results indicate a $\delta^{34}$S range of +12.5 to +14.5 ‰ for that site. Similar values can be expected for the upper Belize Valley based on continuous geological formations spanning from the Petén to western Belize. A slightly higher range of values (+13.6 to +18.8 ‰) for terrestrial faunal remains from Xunantunich ($n = 2$) and Pacbitun ($n = 15$) have been recently reported [49] but are consistent with expected human values who primarily consumed terrestrial resources. Based on these data, we created a model presented in Fig 2 using inverse

**Table 2. Bedrock geology of the study area zones following Cornec (2010).**

| Geologic Age | Eastern lowland Geographic Zone | Associated Archaeological Sites | Catchment Bedrock Geology | Modeled $\delta^{34}$S Values (‰) |
|---|---|---|---|---|
| Late Tertiary | Belize Valley | Baking Pot, Cahal Pech, Lower Dover, Xunantunich/San Lorenzo | Red Bank Group: bentonitic clays | +12 to +16 |
| Early Tertiary | Belize Valley Southern Periphery | Barton Creek, Pacbitun, Pook's Hill, Tipu | El Cayo/Dubloon Bank formation: limestones with chert and dolomite inclusions | +10 to +14 |
| Cretaceous | Vaca Plateau | Caledonia, Caracol | Undifferentiated limestones, dolomite, collapsed breccias | +10 |
| Permian | Maya Mountains (incl. Mountain Pine Ridge) | Peligroso, Ramonal | Santa Rosa Group: argillaceous sediments including shales, phyllites, and carbonaceous argillites | > +8 |

Modeled $\delta^{34}$S isoscape values in human bone collagen depicted in Fig 2.

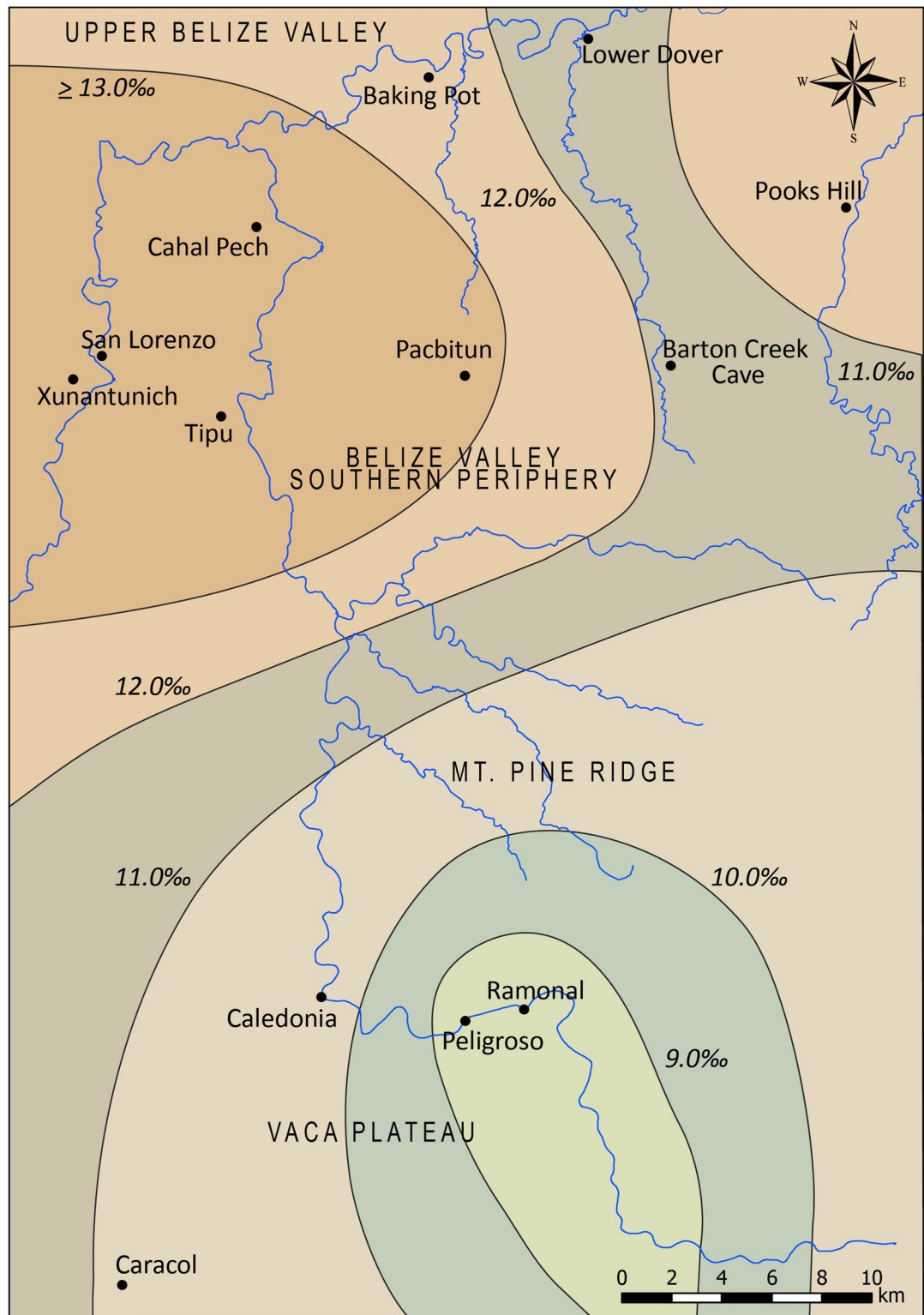

**Fig 2. Modeled δ³⁴S isoscape for the Eastern Maya lowlands.**

distance weighted interpolation of expected values and average values by site where available (Xunantunich, Pacbitun). Analyses were performed using the ArcMap$^{TM}$ 8.0 program by Esri, with resulting contours representing predicted $\delta^{34}$S isoscape values.

## Archaeological contexts

A total of 114 human individuals representing 12 archaeological sites from the Eastern Maya lowlands were analyzed for $\delta^{34}$S, $\delta^{13}$C, and $\delta^{15}$N this study (S1 Table). Sites were selected to represent a range of different geological and environmental zones. Burials also span approximately 3000 years of occupational history in the Eastern Maya lowlands, ranging in age from the Late Preclassic to Colonial periods (~300 BCE—1800 CE), allowing us to capture change in diet and mobility through time. Sites included in this study are also those with relatively well-preserved contexts, allowing for $\delta^{34}$S, $\delta^{13}$C, and $\delta^{15}$N measurements of human bone collagen. Temporal assignments were designated for each burial based on AMS $^{14}$C dates from bone collagen when available or relative ceramic dating (S1 Text). Contextual and demographic information are also reported in S1 Table when available in the published literature.

**Baking pot.** Baking Pot is located approximately 10 km east of the modern town of San Ignacio, on the southern bank of the Belize River. Radiocarbon dating of human burials indicates that the site was first occupied at the end of the Middle Preclassic [105–107]. Monumental construction began during the Early Classic and peaked during the Late Classic (600–750 CE), corresponding with the growth of population around the site [105–107]. The site also experienced a Late Postclassic reoccupation, which followed large-scale depopulation associated with the Terminal Classic "collapse" [105, 108].

The burials considered here include those from both the Baking Pot monumental epicenter and peripheral households, all of which have previously reported AMS $^{14}$C dates [see S1 Text]. Two Early Classic burials come from M-209 (Burials 2 and 4) [109], a terminus structure linked by a causeway to the site core. Three epicentral burials were located within peri-abandonment deposits in Plaza B, which consist of large and dense concentrations of ceramics, lithics, faunal materials, and other artifacts [106, 110]. All three Plaza B burials are Late-Terminal Classic in age and were likely placed close to the time of Baking Pot's abandonment [110]. The remaining burials (*n* = 5) are from Baking Pot's peripheral settlement, and their temporal associations span the entire occupational history of the site. Previous $\delta^{13}$C and $\delta^{15}$N measurements of bone collagen for four individuals from the Baking Pot settlement, including the M-102 Burial 1 considered here, demonstrated similar maize-based diets across social classes [109].

**Barton creek cave.** One individual associated with a Terminal Classic context was sampled from Barton Creek Cave, located in the Roaring Creek region of the study area. The cave is named for the creek that flows through it, forming a navigable waterway that runs below several ledges and flowstone-spans that contain cultural material, including pottery and human remains [111]. The remains of the individual considered in this study were placed on the surface of Ledge 11, in the deepest part of the cave that possesses prehistoric material culture [111]. The individual was part of a commingled burial assemblage that was recovered during salvage excavations. No previous isotopic studies have been conducted on individuals from this site.

**Cahal pech.** Cahal Pech was a medium-sized center located in the upper Belize Valley at the confluence of the Macal and Mopan Rivers. The site possesses one of the longest occupational histories in the Maya lowlands [112, 113], spanning from ~1200 cal BCE to 850/900 cal CE, when mortuary activity ceased at the site [114]. This study considers isotopic data from 15 individuals from Cahal Pech that were also sampled from a previous isotopic study of AMS

[14]C dated burials by Ebert and colleagues [6]. Burials were in both site core and peripheral settlement contexts. The site core burials are primarily from the Early and Late Classic periods, including two child burials (Str. B4-1sub and Str. C2 Burial 1) associated with terminal site occupation [114]. AMS [14]C dates from a third child burial from Plaza G (Unit 51, Level 2) dated to the Colonial period, suggest re-visitation to the site after its Terminal Classic abandonment. Strontium isotopic analysis determined that this individual was non-local to the Belize Valley region, with a $^{87}Sr/^{86}Sr$ value that overlaps those of the Petén Lakes region in the Central lowlands [115]. Burials from the Cahal Pech settlement represent three peripheral residential groups. Two individuals from the Tolok Group were associated with a Late Preclassic round structure, along with the interment of several other individuals [116]. Two burials from the Zubin Group are Late Classic in date [117]. A total of five burials were also sampled from the smaller Zotz Group [112] and range from the Late Preclassic to Late Classic in age.

Previous $\delta^{13}C$ and $\delta^{15}N$ measurements of bone collagen from directly dated burials support faunal and palaeobotanical findings, indicating that Preclassic diets were relatively broad compared to Classic period diets, incorporating substantial amounts of both wild and domesticated resources [6, 36, 118, 119]. Maize and other domesticated plants formed important components of diets at Cahal Pech from initial settlement, documented by the presence of maize cupule fragments, squash (*Curcurbita* sp.), and fruits recovered from Preclassic deposits [120, 121]. Excavations in a Middle Preclassic context in the site's periphery yielded a large sample ($n$>25,000) of terrestrial, freshwater, and marine faunal remains representing a diet in which maize was not the only source of protein [36]. While some marine species (e.g., parrot fish) have been documented in this and other Preclassic contexts at Cahal Pech [122], they were likely very minor components or absent in most diets. Previous isotopic analyses also document an increase in maize consumption into the Classic period, with elite diets becoming almost completely reliant on maize-based foods [6].

**Caledonia.** Caledonia is an intermediate elite center located on the western bank of the Macal River on the Vaca Plateau approximately 30 km south of the Belize Valley. Strategically situated at the intersection of the Mountain Pine Ridge and limestone-underlain Chiquibul rainforest, the site's location provided access to a variety of economic and subsistence resources (e.g., freshwater foods from the Macal, game from the Mountain Pine Ridge). Although Caledonia was occupied as early as the Late Preclassic, burials primarily date to the Late and Terminal Classic periods [123]. The 18 individuals in this sample were recovered from four burial contexts. One dates to the Preclassic (Str. C1 Burial 5), one to the transition between the Early and Late Classic (Str. A1 Burial 1), and two date to the Late Classic (Str. C2 Burials 3 and 4) [123]. The three Classic period burials contained the remains of multiple individuals, two of which (Burials 1 and 4) have been interpreted as sequentially used family crypts. Previous $\delta^{13}C$ and $\delta^{15}N$ analyses of human bone collagen and apatite ($\delta^{13}C$ values) identified a maize-based diet supplemented by other plants and animal protein [104]. Stable oxygen isotope analysis of the bone and tooth bioapatite also show that most were local to the Vaca Plateau, although four individuals may have moved to the site from areas with higher $\delta^{18}O$ values, perhaps from the upper Belize Valley [124].

**Caracol.** Caracol, one of the largest Classic period kingdoms in the Maya lowlands, is located approximately 50 km south of the upper Belize Valley in the Chiquibul Forest region of the Vaca Plateau. While there is evidence for Middle and Late Preclassic settlement, Caracol reached its height in the Late Classic with a population of nearly 10,000 people spread over 200 km$^2$ [125]. The 15 individuals in this sample were recovered during conservation efforts by the Tourism Development Project (TDP) from 2000 to 2004, sponsored through the Government of Belize and operated in conjunction with the Institute of Archaeology [126]. All burials sampled come from the Late Classic phases of major elite architectural complexes within the

Caracol site core, including Str. A6 (Temple of the Wooden Lintels), the South Acropolis, and the massive raised architectural complex known as Caana, which served as the site's royal palace [127, 128, see also 125]. Previous $\delta^{13}$C and $\delta^{15}$N analyses of bone collagen from a separate burial assemblage at Caracol, composed of individuals from both the site's epicenter and peripheral settlements, documented a distinct "palace diet" that was high in $C_4$ foods (likely maize) and animal protein, which differed from the diets of the site's non-elite population [129]. Recent $^{87}$Sr/$^{86}$Sr analyses of three individuals from the TDP assemblage indicates that two burials (Str. B26, CD4/B26 and Altar 23 B/1 and 2, CD14) were non-local individuals who migrated to Caracol after their childhood [130].

**Lower dover.**   Lower Dover is located approximately 6 km downriver from Baking Pot where Barton Creek meets the Belize River. Preclassic occupation was concentrated within the site's residential hinterlands [131, 132]. Monumental architecture in the Lower Dover site core was not constructed until the Early to Late Classic periods. No previous isotopic analyses have been published for the site. Two burials from Lower Dover considered for this study come from the site core. The first is a Late Classic burial from Courtyard 1. The second burial was that of an adult male located outside the entrance of a small rockshelter on the northern periphery of the Lower Dover site core, designated as Rockshelter 1 [133]. The individual was placed in a tightly flexed supine position, which is more consistent with Late Postclassic patterns in the upper Belize Valley. Faunal analyses from the site core and peripheral settlement groups reveal heavy dependence on freshwater resources, especially *jute* snails, in the Middle Preclassic [134] that decreased by the Late Classic, suggesting that other terrestrial foods (e.g., maize, animal protein) formed the bulk of Late Classic diets at this site.

**Pacbitun.**   Pacbitun was a medium-sized center in the foothills along the Belize Valley's southern periphery, situated at the juncture of the lowland tropical rainforest and the nearby Mountain Pine Ridge. With initial occupation at the beginning of the Middle Preclassic (~900 BCE), Pacbitun reached its apogee during the Late and early Terminal Classic periods when agricultural terraces were constructed near the site to support its growing population [135, 136]. Stable isotope data presented in this study are from 11 burials. These burials include five from Late and Terminal Classic contexts in Strs. 1, 2, and 5, which together formed an E-Group architectural assemblage in the site core, and one post-abandonment burial in Str. 4 [135]. Five individuals buried in Str. 6, in the southeast quadrant of the site core, and one from Str. 36 in the southwest quadrant, date to the Terminal Classic [137] and are also included in this study.

Previous isotopic analysis of human bone collagen ($\delta^{13}$C and $\delta^{15}$N) found that maize consumption at Pacbitun peaked during the Late Classic, but declined slightly during the Terminal Classic [13, 138]. High status individuals consumed more $C_4$ protein than lower status community members, whose burials were located farther from the epicenter [13]. Other paleodietary data come from zooarchaeological analyses, which indicate that freshwater fauna may have been an important food source during the Preclassic and Early Classic periods. Larger mammalian species, including peccary, brocket deer, and white-tailed deer became preferred foods during the Late and Terminal Classic [139]. While marine fauna have been found in contexts from all time periods at Pacbitun, indicating the site's participation in long-distance trade networks, these resources are unlikely to have been significant components in local diets.

**Pook's hill.**   Pook's Hill is a formal residential *plazuela* group located in the foothills of the Roaring Creek Valley, to the east of the upper Belize Valley. Excavations documented Late and Terminal Classic phase architecture and deposits associated with the site's final occupation [140, 141]. The burials considered for isotopic dietary analyses here include those from Strs. 2A and 4A, and date to the Late Classic. On the eastern side of the *plazuela*, Str. 4A functioned as an ancestor shrine and contained most of the site's burials, including a multiple interment

sampled here (Burial 4A/3 [142]), whereas Str. 2A possibly functioned as a gathering space for the local corporate group as well as foreign visitors [142]. Strontium isotope analyses indicate that all individuals from Str. 4A were local to the Roaring Creek region [143]—with one possible exception from Str. 2A (Burial 2A-1 [see 142]). Although strontium data are not available for that individual, given its contextual association and as the only burial with jadeite inlays from the site, Burial 2A-1 may represent a non-local individual, pending eventual corroboration. Paleoethnobotanical analyses from Pook's Hill documented the presence of domesticates including maize kernels, squash seeds, chili pepper seeds (*Capsicum annuum*), as well as tree fruits such as calabash (*Crescentia cujete*) were part of Late Classic diets [144], which is consistent with previous stable isotopic analyses [143]. Other terrestrial food resources included white-tailed deer, peccary, rabbit, armadillo, and turkey [145]. High frequencies of freshwater shells and turtle remains, as well as marine species such as parrot fish were also present [145], suggesting a mixed terrestrial/freshwater diet for the residents of Pook's Hill with a relatively small input from marine ecosystems.

**Maya mountains sites: Ramonal and peligroso.**   The sites of Ramonal and Peligroso are in the Mountain Pine Ridge region of the Maya Mountains along the Upper Macal River [146]. Both sites were investigated during archaeological mitigation related to the construction of the Macal River Upstream Storage Facility, known locally as the Chalillo Dam. Though these sites are presently underwater, analysis of burials and associated artifacts from salvage excavations prior to dam construction indicates Late Classic occupation. Ramonal was the largest known settlement upstream from the Chalillo Dam site and consisted of three formal patio groups with several peripheral house mounds [146]. Tomb 3 contained the remains of a single individual of indeterminate age and sex (Burial 10). The complexity of the two-chambered tomb, which was constructed from large slate slabs and possessed a "chimney entrance" [146], suggests that the interred individual was of high status. Peligroso is classified as a large *plazuela* group and consisted of five structures. Burial 7, located in the building's eastern shrine structure, contained the remains of a single individual placed in a seated position with arms bound the back, facing east [146].

**Tipu.**   Tipu is located south of the Belize Valley along the Macal River in the foothills of the Maya Mountains. The site's earliest occupation was during the Preclassic when a small farming community was present. By the Late Postclassic, the site had grown into an independent district capital [147, 148]. The most intensive period of occupation occurred during the Colonial period between ~1500–1700 CE, when Tipu was the location of Spanish *visita* mission church. Over 600 burials have been documented in association with the church [147]. Of the 21 individuals from Tipu considered in this study, 10 are Colonial in date (Graham 2011). The remainder have been directly dated to the Postclassic (*n* = 10) and Classic (*n* = 1) (J. Hoggarth, personal communication). Analysis of faunal remains suggests that a diversity of terrestrial species was consumed at Tipu from the Postclassic through Colonial periods, with a special focus on mammals such as armadillo, agouti, and white-tailed deer [149]. Previous stable isotopes analyses of both bone collagen and dental calculus indicate that this was supplemented by a diet with high contributions of maize throughout the site's occupational history [150].

**Xunantunich and san lorenzo.**   Xunantunich was a major center that overlooked the Mopan River before its confluence with the Macal River. Although first settled in the Early and Middle Preclassic periods (~1200–900 BCE), following a brief occupational hiatus the site was re-established as the capital of a large Late Classic Belize Valley polity [151–155]. All burials in this study [previously published, see 156] derive from the site's monumental core except for one dating to the Late Classic from San Lorenzo (Op. 243 LL/3), a rural community in the Xunantunich hinterland that was economically and socially tied Xunantunich during this time [157].

During salvage excavations in the 1980s of Str. B5, located in an elite residential courtyard, a femur fragment (XN Individual 2) from a looters trench dating to the Early Classic, as well as a Late Classic burial (Str. B5 Burial 1) were encountered [158]. Samples also include two burials excavated by the Xunantunich Archaeological Project (XAP) from Group D, of which Op. 74R dates to the Late Classic and Op. 21C Individual 1 dates to the Late Preclassic period [159, 160]. Another burial (Op. 302G) from Str. A-11 was likely associated with termination and abandonment of the building during the Late Classic [161]. The remains of a Late Classic individual were also recovered during excavations of Str. E3 (Op. 4b-24) by the Mopan Valley Preclassic Project (MVPP) in 2008 [162]. Finally, excavations of Str. A9 encountered a vaulted tomb containing the burial of an adult female (Burial 2) [163, 164]. Analysis of AMS [14]C dates in combination with epigraphic data indicates that the burial was placed between 721–775 cal CE [164].

Like other Classic Maya individuals, $C_4$-based diets have been previously noted for individuals from Xunantunich and San Lorenzo, though they were supplemented with $C_3$ plant foods and animal protein [19]. Strontium analyses identified that some of the wild game (e.g., deer and peccary) in the Xunantunich assemblage were not locally hunted, showing acquisition of non-local foods perhaps from the Maya Mountains [165, 166]. Movement was not, however, restricted to dietary resources, as strontium and oxygen isotope ratios suggest that some individuals from both Xunantunich and San Lorenzo moved to their place of burial from isotopically distinct areas outside of the Belize Valley [19, 167].

## Materials and methods

### Sample considerations and treatment

All necessary permits for research in Belize and permissions to export and conduct isotopic analyses of human burials were issued by the Belize Institute of Archaeology, National Institute of Culture and History. This study has complied with all relevant regulations. Considerations were made to judge the preservation of the samples prior to analysis to limit destruction. The sampling strategies, therefore, focused on sampling non-diagnostic elements of bones to preserve the remaining material for future research projects, curation, and with ethical considerations for the post-mortem treatment of human remains [see 168]. Rigorous analytic sampling techniques were used to limit the required material needed to perform the analyses (ultrafiltration of cortical bone) and the failure rates involved in sample preparation [S2 Text].

### Methods

Human skeletal samples were analyzed using standard procedures for collagen extraction following the modified Longin [169] method with ultrafiltration [170; see also 23, 171]. Approximately 100–1000 mg of dry bone from each sample were cleaned, demineralized, ultrafiltered, and lyophilized (i.e., freeze dried). Ultrafiltered collagen was then weighed to determine percent yield as a first evaluation of the degree of bone collagen preservation, and suitable samples were selected for isotopic measurement. A detailed description of these steps is provided in S2 Text. Measurements for $\delta^{13}C$, $\delta^{15}N$, and $\delta^{34}S$ were performed on aliquots of the same bone sample.

Given the collaborative nature of this study, measurements were performed at four different laboratories according to standard procedures [S2 Text]. Analytical uncertainty was calculated according to Szpak and colleagues [172] and ranged from ±0.11 to ±0.44 ‰ for $\delta^{13}C$ values, from ±0.08 to ±0.40 ‰ for $\delta^{15}N$ values, and from ±0.32 to ±1.24‰ for $\delta^{34}S$ values, depending on the laboratory. Despite differences in sample analysis, $\delta^{13}C$ and $\delta^{15}N$ values analyzed by different laboratories are comparable since bone collagen samples are extracted, prepared, and

analyzed using similar methods [173]. Preliminary evidence also indicates that the $\delta^{34}$S values of samples analyzed at different laboratories are comparable [S2 Text].

Quality for all $\delta^{13}$C and $\delta^{15}$N samples was evaluated by % crude gelatin yield, %C, %N, and C:N ratios. C:N ratios for all samples except one (MARC2571) fell between 3.10 and 3.50, indicating good collagen preservation [174, 175]. Because of a high C:N ratio of 3.70, MARC2571 was excluded from further analysis. Quality for $\delta^{34}$S samples was evaluated following Nehlich and Richards [23]. Two samples with C:S ratios that fell well beyond the 600±300 range did not meet quality control standards (BKP15 and TP005) and are therefore not considered in subsequent analyses. An additional three samples had N:S ratios slightly beyond the 200±100 range (BKP47, BKP41, and TP492), and therefore were also excluded from additional analyses. Four samples (BKP09, BKP11; BKP29, BKP35) had a wt % of sulfur exceeding the range from 0.15–0.35% for mammal bones recommended by Nehlich and Richards [23], and therefore were also excluded from further analysis. To avoid including the same individual twice in statistical analyses, the isotopic values from four mandibles that may belong to the same individuals as the seven fibulae from Caledonia Str. C2 Burial 4 (MARC 2525-MARC 2528) were also omitted [see S1 Table for these data].

## Results

The following section presents the stable isotope results that met quality control standards for a total of 101 individuals from 12 archaeological sites that span the Late Preclassic through Colonial periods. Below we offer interpretations about our results within the context of what is known about Maya diet and the local ecology of the Eastern Maya lowlands, in addition to addressing our hypotheses. Summary statistics for $\delta^{13}$C, $\delta^{15}$N, and $\delta^{34}$S values from human bone collagen analyzed in this study are presented by site in Table 3, and the complete dataset is provided in S1 Table, along with relevant citations for isotopic data. Complete results of statistical analyses are reported in the Supplementary Information and are summarized below.

### Carbon and nitrogen isotope results

Carbon and nitrogen isotopic values exhibited a high degree of overlap for individuals from all sites, for all sub-regions of the Eastern lowlands, and for all time periods examined (Fig 3). The $\delta^{13}$C values ranged between –13.4 and –6.9 ‰ and fell along a spectrum consistent with variation for Maya populations consuming 50–70% dietary protein derived from $C_4$ sources [e.g. 8,

**Table 3. Mean $\delta^{13}$C, $\delta^{15}$N, and $\delta^{34}$S values by site for Eastern Maya lowlands.**

| Site | Individuals ($n$) | $\delta^{13}$C (‰ VPDB) | SD | $\delta^{15}$N (‰ Atm $N_2$) | SD | $\delta^{34}$S (‰ VCDT) | SD |
|---|---|---|---|---|---|---|---|
| Baking Pot | 3 | −12.6 | 0.9 | +9.7 | 1.8 | +12.6 | 3.1 |
| Barton Creek | 1 | −9.0 | | +9.3 | | +12.2 | |
| Cahal Pech | 15 | −10.2 | 1.5 | +8.8 | 1.0 | +13.8 | 1.0 |
| Caledonia | 18 | −10.1 | 1.9 | +9.0 | 1.1 | +11.0 | 2.0 |
| Caracol | 15 | −9.6 | 1.2 | +9.4 | 1.3 | +10.3 | 3.8 |
| Lower Dover | 2 | −10.7 | 1.7 | +9.7 | 1.3 | +10.5 | 0.8 |
| Maya Mts. Sites | 2 | −10.7 | 0.5 | +9.6 | 1.1 | +8.5 | 0.0 |
| Pacbitun | 10 | −10.2 | 1.9 | +8.8 | 0.8 | +12.6 | 0.9 |
| Pook's Hill | 6 | −11.4 | 0.7 | +8.4 | 0.3 | +10.0 | 0.7 |
| Tipu | 21 | −9.8 | 1.2 | +9.2 | 0.7 | +12.7 | 1.2 |
| Xunantunich/San Lorenzo | 8 | −11.0 | 1.6 | +9.5 | 0.9 | +13.7 | 1.4 |

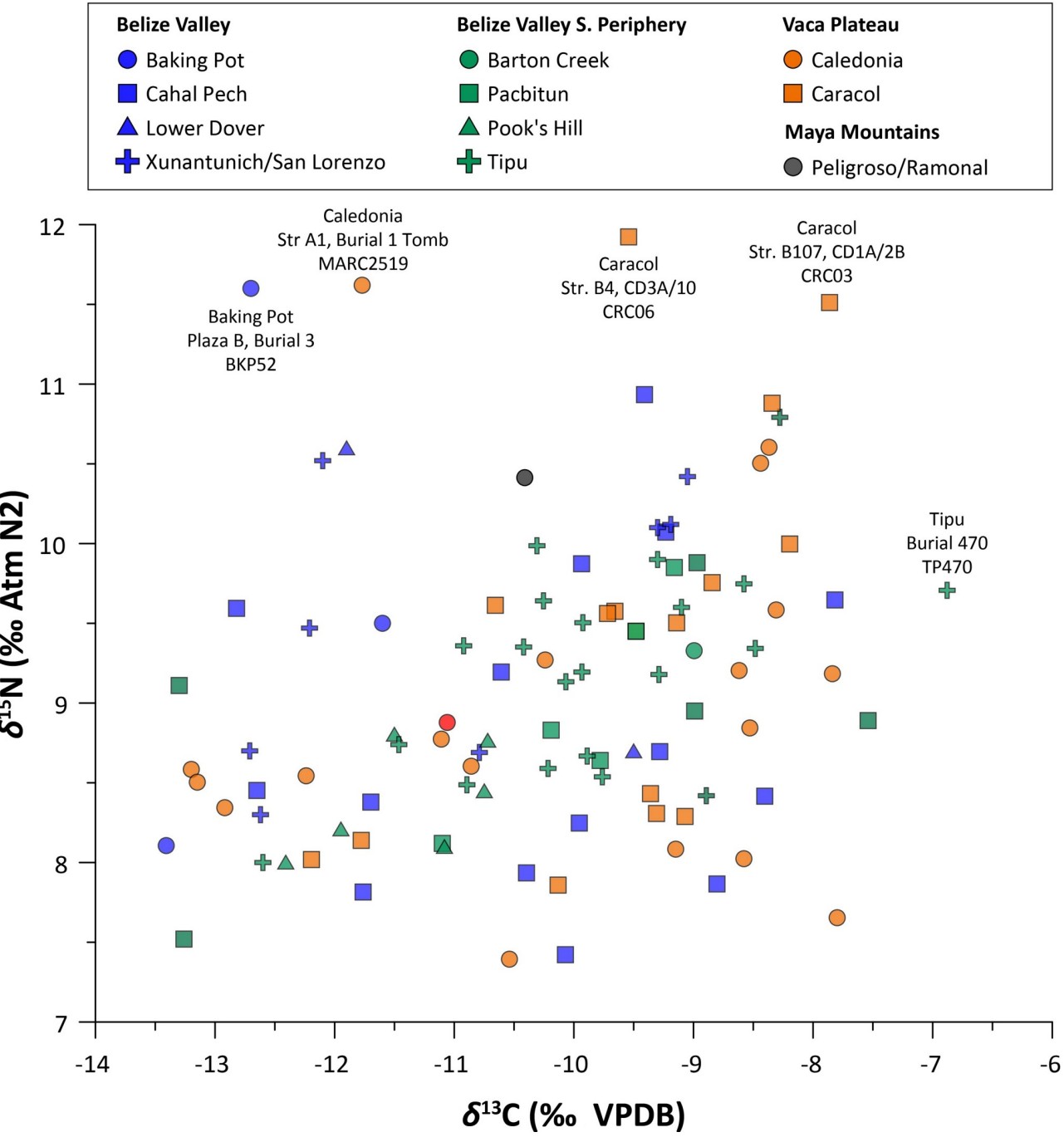

**Fig 3. Eastern lowland human $\delta^{13}$C and $\delta^{15}$N values, with outliers described in text indicated.**

17, 19]. To examine variation between geographic zones, we divided sites by their location in the Eastern lowlands in the upper Belize Valley (*n* = 28), the Belize Valley Southern Periphery (*n* = 38), or the Vaca Plateau zones (*n* = 33; following Table 2). No statistically significant differences were identified between the $\delta^{13}$C values (ANOVA, F = 2.24, *df* = 2, *p* = .112) and $\delta^{15}$N values (ANOVA, F = .42, *df* = 2, *p* = .652) by location. Though we could not include the Maya Mountains in these calculations, because of the small sample size of burials (*n* = 2), a lack of

difference between other zones suggests that C4 plant and animal protein contributions to the diet were relatively similar across the greater Eastern lowlands region.

Comparisons of $\delta^{13}$C values through time (Fig 4 and Table 4) showed a significant increase from the Preclassic to Early Classic period ($t$ = -3.80; $df$ = 7; $p$ = .003), from the Late to Terminal Classic ($t$ = 2.03; $df$ = 18; $p < 0.05$), and again from the Terminal Classic to Postclassic ($t$ = -2.37; $df$ = 22; $p < 0.05$). Higher inputs of C4-foods in the Classic compared to the Preclassic has been documented at other lowland Maya sites (e.g., Altar de Sacrificios and Dos Pilas, [35]; Altun Ha, [14]; Pacbitun, [13]). Dietary shifts from the Terminal Classic to the Postclassic have not been explored in depth for the lowlands except for parts of northern Belize with access to coastal resources [176]. Though most Eastern lowland sites in this study displayed a range of values, Pook's Hill and the Maya Mountains sites (Peligroso and Ramonal) exhibited the tightest range of $\delta^{13}$C values, which can likely be attributed to sampling biases. The relatively small sample sizes for both sites date solely to the Late Classic, preventing examination of fluctuating C4 inputs through time. Individual $\delta^{15}$N values ranged between +7.4 and +11.9 ‰, spanning between one and two trophic levels (+3 to +5 ‰), perhaps suggesting differential access to animal protein or alternatively impacts from *milpa* agriculture and burning. Lower values likely

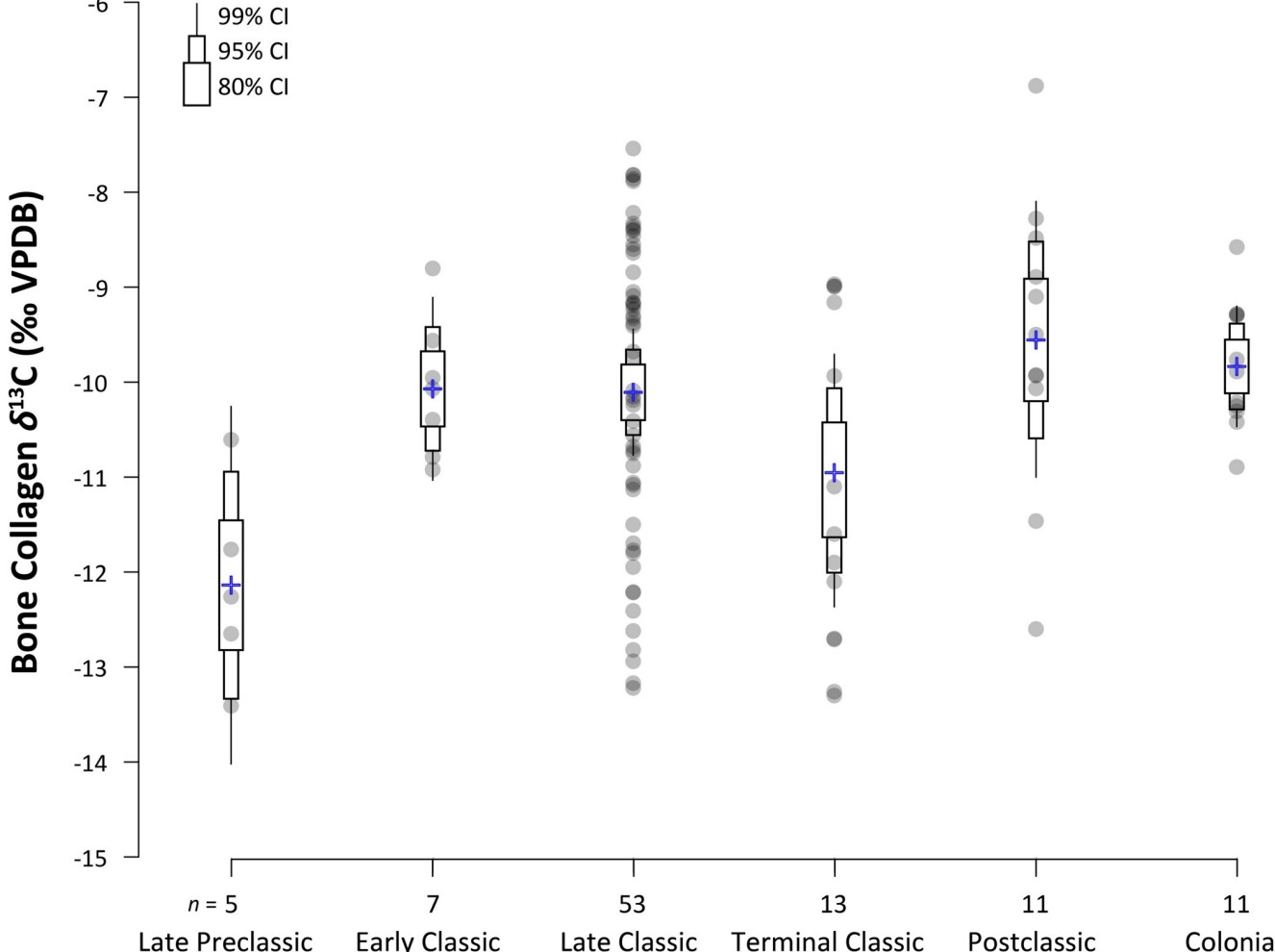

**Fig 4. Bullet graph of stable carbon ($\delta^{13}$C) isotope values for human bone collagen.** Data points are shown as circles with the means for each time period indicated by a cross. The bullet graph shows the 80%, 95%, and 99% confidence intervals (CIs; thickest to thinnest bullets) around the mean.

**Table 4. Mean isotopic values for human bone collagen by time period.**

| Time Period | $n$ | $\delta^{13}C$ (‰ VPDB) | SD | $\delta^{15}N$ (‰ Atm $N_2$) | SD | $\delta^{34}S$ (‰ VCDT) | SD |
|---|---|---|---|---|---|---|---|
| Late Preclassic | 5 | −12.1 | 1.0 | +8.4 | 0.5 | +13.5 | 1.5 |
| Early Classic | 7 | −10.1 | 0.7 | +8.8 | 1.5 | +13.1 | 1.3 |
| Late Classic | 53 | −10.1 | 1.6 | +9.1 | 1.0 | +11.2 | 2.7 |
| Terminal Classic | 13 | −11.0 | 1.7 | +9.4 | 1.1 | +12.7 | 1.3 |
| Postclassic | 11 | −9.6 | 1.6 | +9.2 | 0.7 | +12.5 | 1.6 |
| Colonial | 11 | −9.8 | 0.7 | +9.2 | 0.6 | +13.1 | 1.4 |

reflect terrestrial protein resources; however, there were no significant temporal differences in $\delta^{15}N$ values.

The $\delta^{15}N$ values of four individuals were identified as statistical outliers for the entire dataset ($\bar{x} \pm 2\sigma$ = +9.1 ± 2.0 ‰). They include at least two subadults, one each from Baking Pot (BKP52) and Caledonia (MARC2519). Two individuals from Caracol also had high $\delta^{15}N$ values, though age information is unavailable (CRC06 and CRC14). When the entire dataset is considered, mean $\delta^{15}N$ values were significantly higher for subadults (including four infants) compared to adults ($t$ = -2.03; $df$ = 12; $p$ = .03). While this pattern corresponds with published $\delta^{15}N$ data demonstrating a positive trophic shift for infants < 3–4 years of age related to the consumption of breast milk [177–179, see also 180], ages are unknown for three of the four outlier individuals. Finally, the highest $\delta^{13}C$ value in the dataset (−6.9 ‰) was associated with an infant from Tipu (Burial 470) dating to the Early Postclassic. Because Tipu had some of the highest $\delta^{13}C$ values for the Eastern lowlands generally, the value was not a statistical outlier ($\bar{x} \pm 2\sigma$ = -10.2 ± 3.2 ‰). Isotopic studies of pre-industrial maize farming populations across the Americas have also documented $\delta^{13}C$ increases in infants, corresponding to the consumption of $C_4$-foods in addition to $^{13}C$-enriched breast milk [178, 181]. Our analyses, however, found no significant differences in $\delta^{13}C$ values between adults and subadults, including infants.

## Sulfur isotope results

Sulfur isotopic values ranged from +7.0 to +16.7 ‰, consistent with expectations based on our isoscape model. When $\delta^{34}S$ data are categorized by site (Fig 5), five statistical outliers were identified (Cahal Pech, CHP24; Caledonia, MARC2534; Xunantunich, MARC4574 and XUN05; Caracol, CRC20). The CRC20 individual from Caracol was an extreme outlier for the entire dataset (−1.6 ‰) that falls well outside the expected range of $\delta^{34}S$ values for Eastern lowland humans [49, 90]. Quality control measures suggest that the value is not influenced by diagenesis and an $\delta^{15}N$ value (+10.0 ‰) does not indicate a reliance on wetland resources which might be reflected by lower $\delta^{34}S$ values. Divergent values in bone collagen more likely relate to origin, or less likely to non-local foods, that may have originated outside of the Maya lowlands. Additional data are needed to evaluate these and other scenarios.

Two distinct, statistically significant groups were identified by comparing mean $\delta^{34}S$ values by site (Figs 6 and 7; see also [S3 Text]). The mean $\delta^{34}S$ values of each group (Table 5) reflect the expected gradient based on geological data, with higher values in the upper Belize Valley in the north (Group 1; range of +9.8 to +16.7 ‰) and the lowest values in the Maya Mountains to the south (+8.5 ‰). While the $\delta^{34}S$ values overlap between groups, they generally conform to the geographic zones described above, with some exceptions. Group 1 is composed of 58 individuals from Belize Valley sites as well as those in the Belize Valley's southern periphery, with a mean value of +13.1 ± 1.3 ‰. Group 2 includes 45 individuals from the Vaca Plateau sites of Caledonia and Caracol in addition to Pook's Hill in the southern periphery and Lower Dover, in the Belize Valley proper. Group 2 has the widest range of $\delta^{34}S$ values (+7.0 to +15.9 ‰) with

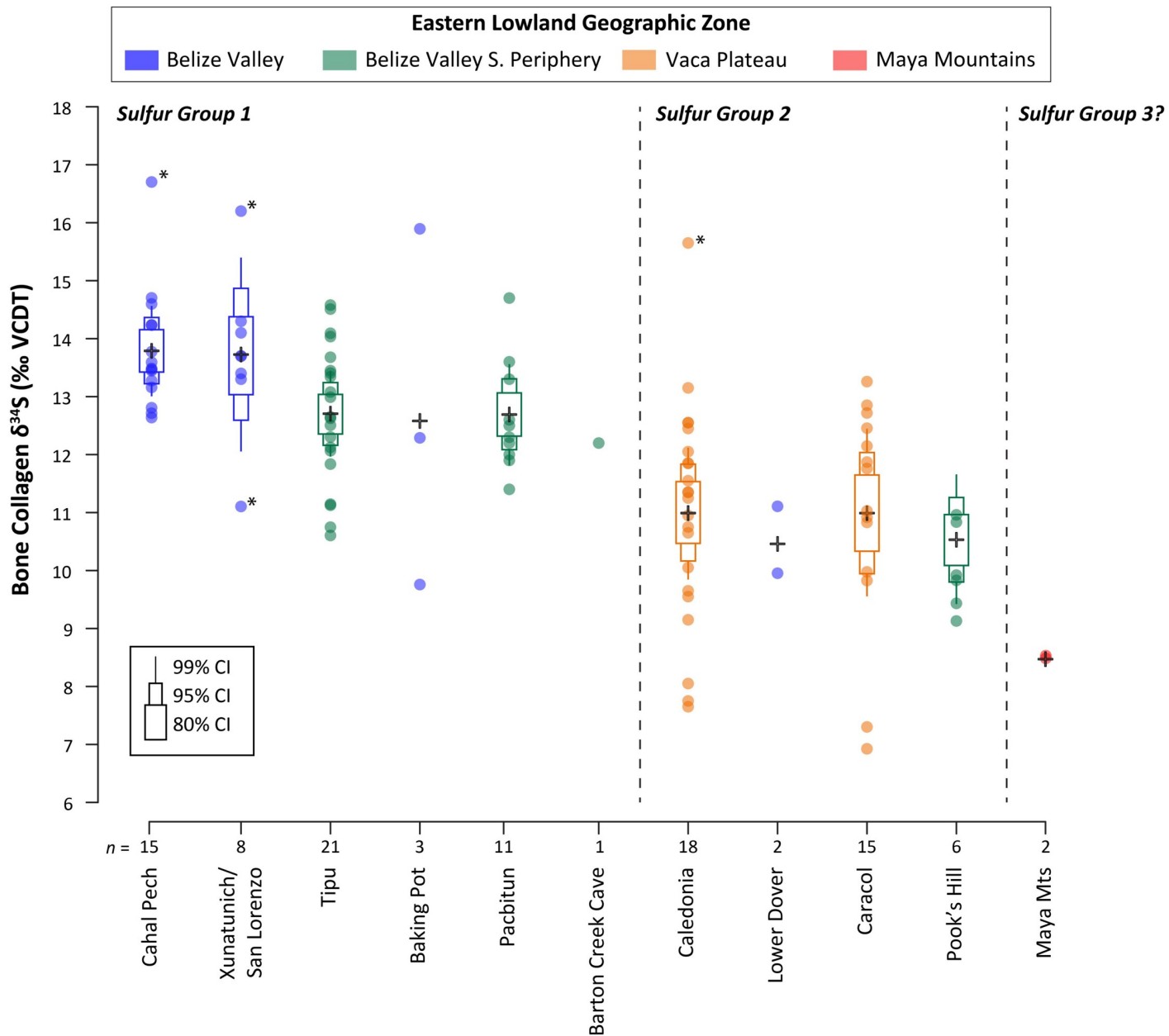

**Fig 5. Plot of the mean $\delta^{34}$S values by site and sulfur group.** Data points are shown as circles with the means for each context indicated by a cross. The bullet graph shows the 80%, 95%, and 99% confidence intervals (CIs; thickest to thinnest bullets) around the mean (cross). Statistical outliers indicated by a star.

a mean value of +10.7 ± 2.6 ‰. This broad range is partly driven by outliers but may also result from different geographic zones within Group 2, the use of isotopically distinct catchment zones at sites like Caledonia [156], or sulfate inputs from streams and rivers draining from the Maya Mountains through different geological zones. A third possible group with mean values that fall outside of the 99% confidence interval around the means of the other two groups consists of the two Mountain Pine Ridge individuals who exhibited the same low $\delta^{34}$S value (+8.5 ‰), possibly indicating a smaller and lower range of values for individuals from the Maya Mountains. Significant temporal differences are also noted in the $\delta^{34}$S values. These changes,

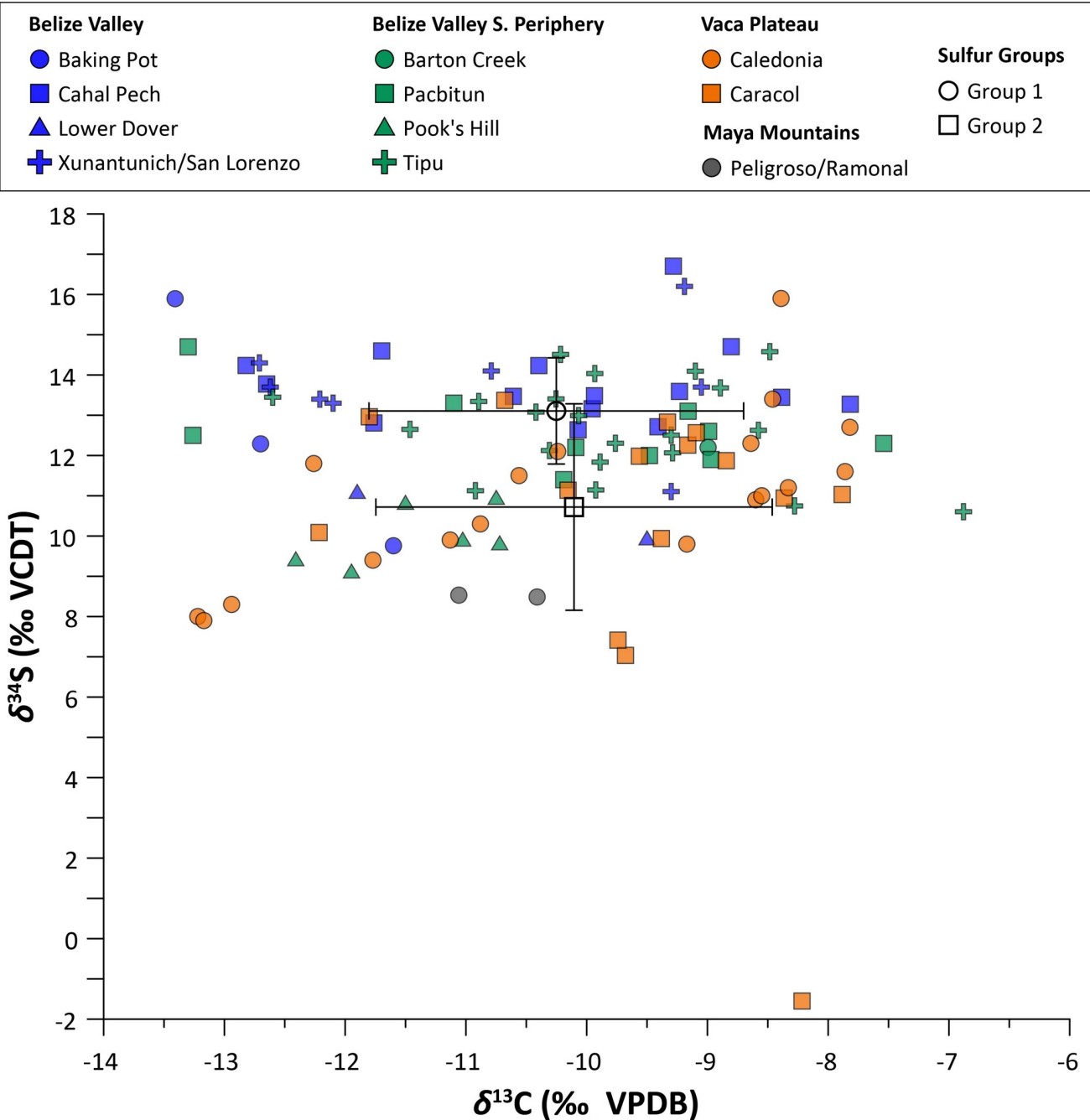

**Fig 6. Eastern lowland human $\delta^{13}$C and $\delta^{34}$S values plotted by geographic zone.** Open symbols and error bars indicate the mean and standard deviation of $\delta^{34}$S values in each sulfur group.

however, are likely the results of sampling bias as they correlate with the predominant geographic zones represented by our sulfur isoscape model in each time period (see [S3 Text]).

Multivariate statistical analyses indicate weak relationships between $\delta^{13}$C and $\delta^{34}$S values and between $\delta^{15}$N and $\delta^{34}$S values at the regional level, as well as at the site level (Figs 6 and 7; see also [S3 Text]). The exception is the site of Caledonia, where there was a strong positive correlation between $\delta^{13}$C and $\delta^{34}$S values ($n = 18$; $r = 0.703$, $p < 0.001$; $r_s = 0.681$, $p < 0.05$).

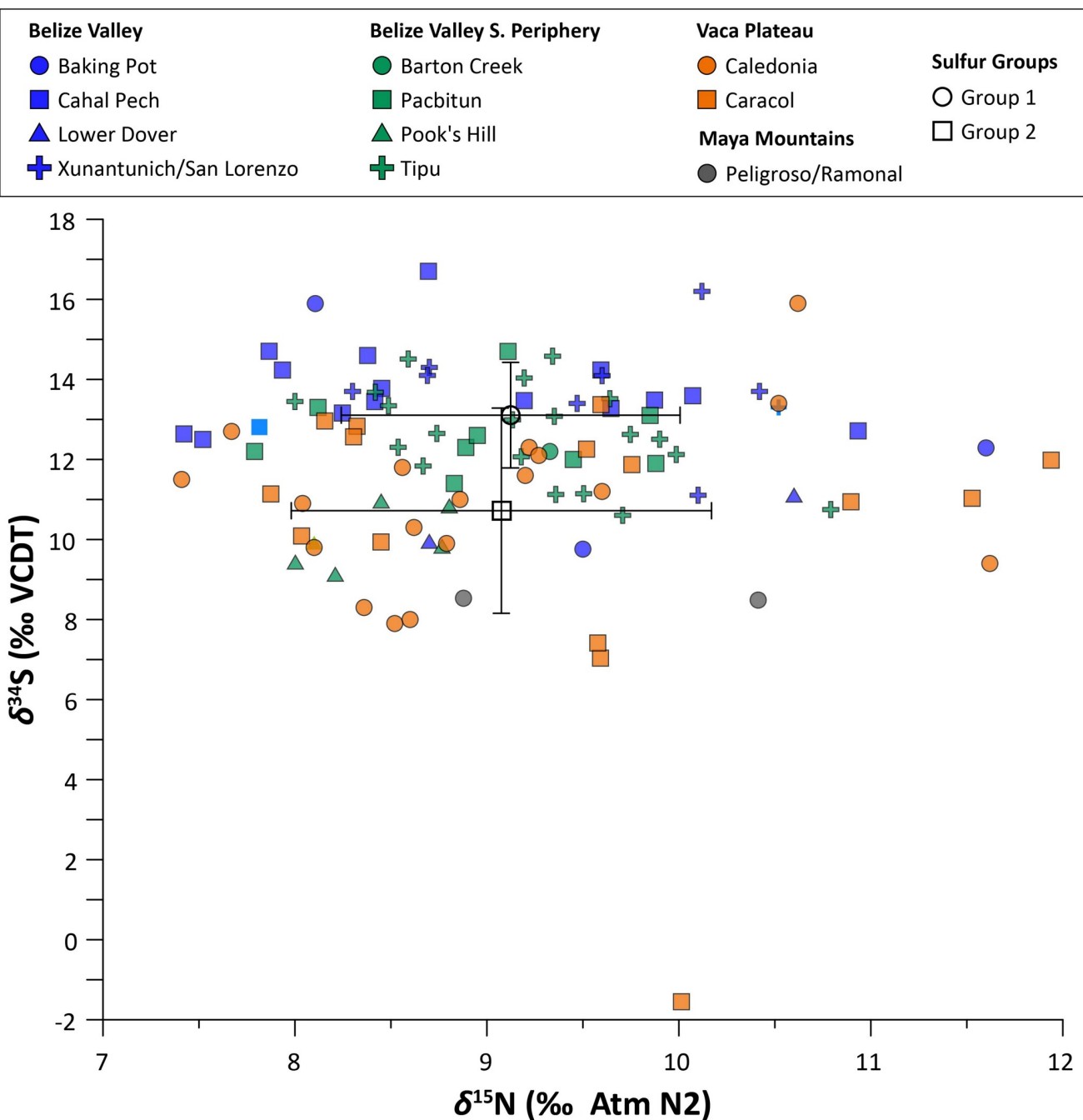

**Fig 7. Eastern lowland human $\delta^{15}$N and $\delta^{34}$S values plotted by geographic zone.** Open symbols and error bars indicate the mean and standard deviation of $\delta^{34}$S values in each sulfur group.

There was also a significant positive correlation between $\delta^{15}$N and $\delta^{34}$S values ($n = 18$; $r = 0.544$, $p < 0.05$) at this site, though the rank-order correlation was moderate (not statistically significant).

No strong or statistically significant correlations between $\delta^{13}$C and $\delta^{34}$S or $\delta^{15}$N and $\delta^{34}$S values were found for any geographic zone of the Eastern lowlands. Correlations between $\delta^{13}$C and $\delta^{34}$S and $\delta^{15}$N and $\delta^{34}$S values were also very weak for both adults and subadults.

**Table 5. Summary statistics for $\delta^{34}$S values by statistically identified sulfur groups for the Eastern Maya lowlands.**

| Sulfur Group | Sites | Indiv. ($n$) | Mean $\delta^{34}$S (‰ VCDT) | SD | Min $\delta^{34}$S (‰ VCDT) | Max $\delta^{34}$S (‰ VCDT) |
|---|---|---|---|---|---|---|
| 1 | Cahal Pech, Xunantunich/San Lorenzo, Baking Pot, Barton Creek Cave, Tipu, Pacbitun | 58 | +13.1 | 1.3 | +9.8 | +16.7 |
| 2 | Caledonia, Caracol, Lower Dover, Pook's Hill | 44 | +11.0 | 1.8 | +7.0 | +15.9 |
| 3 (?) | Maya Mountains Sites (Ramonal and Peligroso) | 2 | +8.5 | 0.0 | +8.5 | +8.5 |

The minimum value for Group 2 does not include the extreme outlier from Caracol.

## Discussion

The application of sulfur isotope analyses ($\delta^{34}$S) to expand our understanding of diet based on carbon ($\delta^{13}$C) and nitrogen ($\delta^{15}$N) isotope analyses from human bone collagen is relatively new in Maya archaeology [49, 66, 90, 118, 156]. Paleodietary studies from elsewhere in the ancient world have successfully demonstrated that $\delta^{34}$S values can be used to identify differences in prehistoric diets related to where food resources come from (e.g., terrestrial, freshwater, and coastal/marine habitats) as well as social factors such as sex and age [21, 22, 24, 182–188]. $\delta^{34}$S data also supplement other isotopic analytical techniques (e.g., strontium and oxygen) by offering insights into the migration and movement of past populations [186].

Though sulfur isotopes have the potential to distinguish between terrestrial, coastal/marine, and freshwater food resources when present, our results found no strong relationships between $\delta^{13}$C and $\delta^{34}$S values or between $\delta^{15}$N and $\delta^{34}$S values for the Eastern lowlands. We argue that the lack of correlation is likely attributable to a relatively stable, homogenous regional diet, with $C_4$ plant foods (maize) forming the main protein component. Paleoethnobotanical data indicate the additional consumption of domesticated $C_3$ plants (beans, squash, manioc [144, 189]), supplemented by wild forest plants and terrestrial animal resources [36, 65]. Isotopic data suggest that this maize-rich diet (>70% of dietary protein) remained relatively constant from the Preclassic to the Colonial period, except for increases in maize consumption demonstrated by positive shifts in $\delta^{13}$C values. The first significant increase occurs from the Preclassic to Classic and the second from the Late to Terminal Classic, corresponding to the appearance of terraces, water management systems, and other large-scale landscape modifications developed to support intensive agriculture, suggesting that maize fueled both population and polity growth [190–194].

The highest $\delta^{13}$C values for the Eastern Maya lowlands were recorded during the Postclassic ($\bar{x} \pm 2\sigma = -9.4 \pm 1.5$ ‰) and Colonial ($\bar{x} \pm 2\sigma = -9.8 \pm 0.7$ ‰) periods. Stable isotopic paleodietary studies of Postclassic populations from northern Belize (e.g., Lamanai, [195]) similarly show a marked increase in maize consumption compared to the Classic, supplemented with costal resources obtained through trade. Our Postclassic sample from western Belize dates primarily to the Late Postclassic, following an occupational hiatus in the Early Postclassic [105]. Additionally, both the Postclassic and Colonial samples are represented almost entirely by individuals from the inland site of Tipu. Changes in burial patterns [107, 196], coupled with $^{87}$Sr/$^{86}$Sr and $\delta^{18}$O isotopes from primary burials, indicate origins for many Late Postclassic individuals outside of the upper Belize Valley region [197]. The significant positive shift in $\delta^{13}$C may reflect changes in economy and settlement with the newly arrived migrants. Archaeological and ethnohistorical sources indicate that maize was the primary staple of Eastern lowland diets during the Late Postclassic and continued to be the most important protein source after Spanish contact [198], although in some cases diets may have expanded to include European domesticates such as cow (*Bos taurus*) and pig (*Sus scrofa*) [199, 200]. High Postclassic

and Colonial $\delta^{13}$C values documented in this study support the stability of a maize-based dietary regime, though it remains unclear to what extent domesticated animals were consumed.

Consistent $\delta^{15}$N values from the Preclassic through the Colonial period, as well as a lack of correlation between $\delta^{15}$N and $\delta^{34}$S values through time, suggests that Eastern lowland communities consumed similar amounts of animal protein across all time periods. Preferred species in zooarchaeological assemblages included locally procured terrestrial herbivores (e.g., deer, rabbit) [65], as well as freshwater snails (*Pachychilus* spp.) and mussels (*Nephronais* spp.) [134, 201, 202], which preliminary evidence suggests have lower $\delta^{15}$N values [35]. There is comparatively little zooarchaeological evidence, however, for the consumption of freshwater fish at Belize Valley sites during the Classic period [65].

A positive shift in mean $\delta^{15}$N values beginning in the Late Classic may instead be related to the intensification of maize production through *milpa* agriculture, which can result in higher $\delta^{15}$N values of plants that are then passed onto consumers [63]. Forest loss, which alters $^{15}$N sources for plants, can also increase the $\delta^{15}$N values of both humans and animals [67, 69, 203]. In the Maya lowlands, forest loss was influenced by not only milpa farming but also other forms of intensive agriculture (e.g., terracing) needed to support large Late Classic populations [194, 204]. Climatic drying and deforestation driven by increased demands for fuel, construction material, and agricultural land associated with large populations were pervasive across the lowlands during the Late and Terminal Classic [204, 205]. These conditions likely impacted the $\delta^{15}$N values of plant and animal food resources, and subsequently, those of humans. Climates also began a drying trend after ~700 CE, resulting in extreme drought conditions persisting through the ninth century Terminal Classic "collapse" [205]. It is possible that these drought conditions impacted the $\delta^{15}$N values of fauna consumed by the Maya, as has been documented in other contexts [67, 69, 206]. Reductions in rainfall have also been correlated with increased $\delta^{15}$N values within a trophic level for animals consuming more than 10% $C_4$ plant protein, which would include Late and Terminal Classic Maya human populations [6, 207].

Sulfur isotope data can help clarify inferences about food sources drawn from $\delta^{13}$C and $\delta^{15}$N data. Although $\delta^{34}$S values of marine and terrestrial fauna overlap at Maya sites, preliminary data suggest terrestrial consumers may exhibit elevated $\delta^{34}$S values relative to those dependent on freshwater resources [49]. Humans reliant on terrestrial, maize-based protein are therefore expected to have higher $\delta^{34}$S and $\delta^{13}$C values and lower $\delta^{15}$N values than those who consumed more protein from freshwater fish or reptiles. The lack of correlation between $\delta^{13}$C, $\delta^{15}$N, and $\delta^{34}$S values in human bone collagen from the Eastern lowlands reflects other lines of archaeological data indicating primarily terrestrial diets that relied heavily on various $C_4$ and $C_3$ protein sources, perhaps with some freshwater protein. Based on our isotopic analyses, there is no strong evidence for significant marine/coastal food consumption for any time period. Instead, when considered alongside $\delta^{34}$S values, consumption of terrestrial meat protein is likely the dominant driver of $\delta^{15}$N values (possibly coupled with the $\delta^{15}$N values of plants mentioned above). This suggests that positive shifts in $\delta^{15}$N values are more likely related to ecological change from the Late Classic to the Postclassic (e.g., drought) [67, 69] rather than different dietary regimes, though higher Colonial values could possibly be related to an increased reliance on animal protein, especially domesticates such as cow and pig, compared to pre-Colonial periods [199].

An exception is noted at the site of Caledonia, where there is a strong positive correlation between $\delta^{13}$C and $\delta^{34}$S values. The site was strategically located on the eastern edge of the Vaca Plateau along the Macal River, granting access to resources from multiple ecological niches in neighboring geographic zones, including the Mountain Pine Ridge region of the Maya Mountains. At Caledonia, individuals with higher $\delta^{13}$C and $\delta^{34}$S values likely consumed maize-based

diets local to the site, as elevated $\delta^{34}$S are reflective of the local limestone geology of the Vaca Plateau. On the other hand, individuals with lower $\delta^{13}$C and $\delta^{34}$S values were probably more reliant on protein from surrounding regions, including freshwater snails and mussels from the Macal River or animals from the Mountain Pine Ridge, which has a lower geological $\delta^{34}$S signature [156]. The Caledonia dataset illustrates that the inclusion of $\delta^{34}$S data in paleodietary studies can reveal how people in the past strategically accessed localized and isotopically distinct micro-environments.

Stable sulfur isotopic data can also indicate whether individuals were recent migrants to their places of burial. Maya archaeologists have typically applied strontium ($^{87}$Sr/$^{86}$Sr) bone chemistry analyses to identify nonlocal individuals since, like $\delta^{34}$S values, $^{87}$Sr/$^{86}$Sr values reflect those of the underlying geology [130, 208, 209]. Migration from isotopically distinct regions is inferred when $^{87}$Sr/$^{86}$Sr values in tooth enamel, which forms during infancy and childhood, differ from the geologic $^{87}$Sr/$^{86}$Sr values of the burial location [210]. At the outset of our study, we hypothesized that different geographic zones of the Eastern lowlands would exhibit distinct $\delta^{34}$S values because of wide variations in geology, and our results largely conform to this assumption about the sulfur isoscape model. Comparisons between human tooth enamel $^{87}$Sr/$^{86}$Sr data and the $\delta^{34}$S values of bone collagen are also useful for examining patterns of migration and mobility across an individual's lifetime in regions, such as the Eastern lowlands, where people had relatively homogenous diets. While previously published $^{87}$Sr/$^{86}$Sr data were not available for all individuals sampled in this study (S2 Table), comparisons demonstrate weak correlations between the "local" $\delta^{34}$S and $^{87}$Sr/$^{86}$Sr values from tooth enamel within sulfur Group 1 (Fig 8), which suggests most individuals lived the last several years of their lives in the same region of the Eastern lowlands in which they were born.

Outliers in the $\delta^{34}$S dataset were similarly identified as non-local individuals by their $^{87}$Sr/$^{86}$Sr values. For example, two individuals in Group 1 have nonlocal $^{87}$Sr/$^{86}$Sr values consistent with those in the Central lowlands (~0.7076) [115, 130, 167]. They also have much higher $\delta^{34}$S compared to other Group 1 individuals (+16.2 to +16.7 ‰), which more closely align with the expected range of $\delta^{34}$S values for humans from the Central lowlands [66, 156]. Because bone remodels over a period between 5–10 years, their nonlocal $\delta^{34}$S values indicate their collagen had not yet equilibrated with the isotopic values of the Eastern lowlands, suggesting that they were more recent migrants to the area. At least one of these individuals who was buried at Cahal Pech (Plaza G, Unit 51) may represent Colonial-period movement from Spanish missions around Lake Peten Itzá (e.g., Mission San Bernabé) [211]. Finally, one individual from Group 2 with low $\delta^{34}$S values has $^{87}$Sr/$^{86}$Sr values consistent with individuals from Southern Belize and the Southern lowlands. While no $\delta^{34}$S data from is yet available for these regions, these comparisons will allow us to predict $\delta^{34}$S values and expand our sulfur isoscape model to other parts of the Maya region.

Conversely, some individuals have non-local $^{87}$Sr/$^{86}$Sr but their $\delta^{34}$S values fall within the expected local ranges for their place of burial. For example, three individuals from sulfur Group 1 had "local" $\delta^{34}$S values but non-local $^{87}$Sr/$^{86}$Sr values of ~0.7092 (Fig 8), indicating an origin south of the Belize Valley in the vicinity of the Maya Mountains [19, 118, 167]. Three individuals with even higher $^{87}$Sr/$^{86}$Sr values, indicate they spent their childhoods in the Maya Mountains [19] or where alluvial sediments washed down from the mountains along waterways into the foothills [130]. An extreme outlier (Tipu, Burial 92) in the non-local sample from sulfur Group 1 ($^{87}$Sr/$^{86}$Sr = 0.712650) [197] may also be from a different part of the Maya Mountains, but has a $\delta^{34}$S value (+14.1 ‰) consistent with Belize Valley populations. The "local" $\delta^{34}$S values (i.e., values reflecting geology at the burial location) are unlikely to be explained by equifinality, whereby different geological areas of the Maya lowlands have overlapping isotopic values, because both $^{87}$Sr/$^{86}$Sr and $\delta^{34}$S values are both derived from the

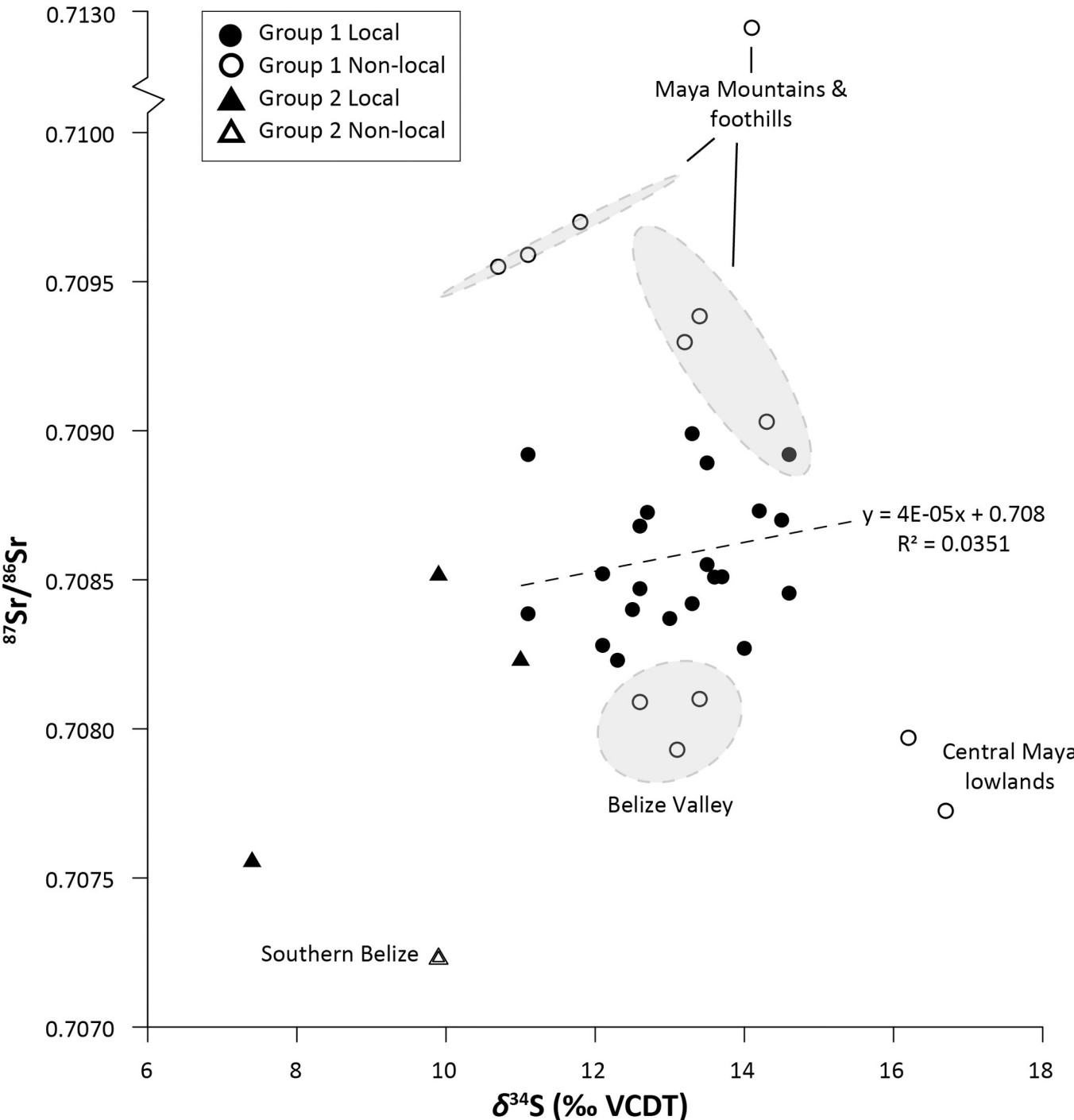

**Fig 8. Comparison of human $^{87}$Sr/$^{86}$Sr and $\delta^{34}$S values for the Eastern Maya lowlands.** Correlations between $^{87}$Sr/$^{86}$Sr and $\delta^{34}$S and regression lines were calculated only for individuals only identified as "local" in previous strontium-based studies from sample sizes $n > 5$ (Group 1). Possible origins for non-local individuals are proposed based on published $^{87}$Sr/$^{86}$Sr data (see S2 Table).

underlying geology and should indicate similar places of origin. Instead, it is probable that these individuals were born elsewhere and had lived in the region associated with their sulfur group areas long enough for their bone collagen $\delta^{34}$S values to equilibrate with those of the

local environment. These preliminary comparisons suggest that $\delta^{34}$S data may be extremely useful for life human history studies related to mobility and migration.

## Conclusions

Applications of stable sulfur isotopic analyses to prehistoric dietary reconstruction has grown over the past decade, but has only recently been applied in Mesoamerica or the Maya lowlands. This study represents one of the largest paleodietary studies of prehistoric human populations to use this technique, allowing us to examine both environmental and social factors that impacted $\delta^{34}$S values in the Maya lowlands. We focused on the types of information that $\delta^{34}$S data can provide about where people from the Eastern lowlands procured their food and what types of foods they consumed, as well as identifying migration in conjunction with other isotopic assays. A long-term perspective, granted by samples that span almost 2000 years of Maya prehistory, suggests that a multi-isotopic approach that includes sulfur isotope analysis offers potential for reconstructing foodways and can provide detailed understandings of movement in the Maya lowlands.

Based on our results, we draw two primary conclusions. First, $\delta^{34}$S values of human bone collagen correspond to expectations about ancient Maya diet developed using other lines of archaeological evidence (zooarchaeological, paleoethnobotanical). In the Eastern lowlands, which is located approximately 80 km west of the Caribbean coast, maize and other terrestrial resources, possibly supplemented with freshwater foods, dominated regional foodways. The general lack of correlation between $\delta^{13}$C, $\delta^{15}$N, $\delta^{34}$S values reflects these terrestrially based diets and implies that diets were generally stable through time except for increases in maize consumption, as documented through $\delta^{13}$C values. Nevertheless, we point out that as the first large-scale study of human $\delta^{34}$S from the Maya lowlands, our interpretations of diet are preliminary, and further research is necessary to understand how $\delta^{34}$S values reflect foodwebs in our study area. Because systematically collected faunal baseline data are absent for most of the Eastern lowland sites discussed here, we are not able to confidently assess the inputs of freshwater resources. Additional baseline $\delta^{34}$S data from food resources will help to better illustrate differences between both human and animal diets. Furthermore, this study focused on a relatively localized area where humans had a homogenous diet. Future work should concentrate on comparisons with other parts of the Maya lowlands where diets would been influenced by higher coastal and marine foods.

Second, $\delta^{34}$S values from human bone collagen reflect distinctive geographic zones in the Eastern lowlands, which negatively correlate with the age of the bedrock geology local to each site. In the Eastern lowlands, bone collagen $\delta^{34}$S values that fall within expected local ranges reflect local habitation for at least the last 5–10 years of an individual's life. In contrast, $\delta^{34}$S signatures that significantly deviate from expected means can potentially identify individuals who lived at the location where they were buried for a shorter length of time. Because of the relationship between $\delta^{34}$S values and geology, our results suggest a promising use for sulfur analyses to track human mobility and migration patterns in Mesoamerica. When considered alongside $\delta^{13}$C and $\delta^{15}$N data, $\delta^{34}$S values that deviate from expectations based on location may also potentially identify the consumption of food resources from other geological zones. Correlations between $\delta^{13}$C and $\delta^{34}$S values at the site of Caledonia, for example, are likely related to resource procurement from multiple, isotopically distinct ecological/geological zones.

Though stable sulfur isotope analysis has not yet been widely applied in Mesoamerica, the results of our research suggest the potential for a multi-isotopic interpretive framework that can take advantage of the dual role of sulfur isotopes as both a dietary and provenance proxy.

At the macro-scale, $\delta^{34}$S data reflect differences in regional ecology and resource exploitation. For example, future baseline studies will provide better understandings of the differences in $\delta^{34}$S values among terrestrial, freshwater, and marine food resources consumed by humans in the Maya lowlands, contributing to interpretations of the ways in which local resources supplemented maize agriculture. Because stable isotope data are specific to an individual, they also lend themselves to exploring socially mediated dietary choices. When used in conjunction with other dietary proxies such as $\delta^{13}$C and $\delta^{15}$N datasets, $\delta^{34}$S values could help to eliminate potential foods from individual dietary regimes, indicating dietary differences based on social or economic factors within and between communities. Similar factors might also influence individual movement or migration. When combined with other indicators of mobility, like strontium isotopes from tooth enamel that document origin at birth, sulfur isotopic analysis is a useful method for examining migration across a person's lifespan. Regionally specific studies in the Maya lowlands represent an initial step to understand variation in isotope ecologies at larger scales in the lowlands, and across Mesoamerica more broadly. Future work comparing sulfur isotopic systems between different lowland sub-regions at different time periods will help clarify the role of ancient Maya diets and migration in both environmental adaptions and social dynamics through time.

## Supporting information

**S1 Table. Stable isotope data from the Eastern Maya lowlands.** Samples highlighted in red failed to meet quality control standards (following Nehlich and Richards 2009) and are not considered for statistical analyses.
(XLSX)

**S2 Table. Published strontium ($^{87}$Sr/$^{86}$Sr) values and associated origins for Eastern lowland Maya individuals included in this study.** Note that not all samples considered here have previously been subjected to $^{87}$Sr/$^{86}$Sr analyses.
(XLSX)

**S1 Text. Temporal assignments for directly dated burials.**
(DOCX)

**S2 Text. Stable isotope sample preparation and measurement methods.**
(DOCX)

**S3 Text. Statistical analyses of $\delta^{13}$C, $\delta^{15}$N, and $\delta^{34}$S data.**
(DOCX)

## Acknowledgments

We would like to thank the Belize Institute of Archaeology, directed by Dr. John Morris, for permission to export and conduct analyses on materials from Belize. Dr. Paul Healy and the Department of Anthropology of Trent University provided the Caledonia and Pacbitun samples for analysis. We also show our gratitude to Alison Pye (CREAIT TERRA Facilities Stable Isotope Laboratory, Memorial University), Anthony Faiia (Stable Isotope Laboratory, University of Tennessee Knoxville), and Paul Middlestead (Ján Veizer Stable Isotope Lab, University of Ottawa) who analyzed the Caledonia, Pacbitun, most Xunantunich, and San Lorenzo samples, and Dr. Jelmer Eerkens and the UC Davis Archaeometry (Stable Isotope) Lab for assisting with early pilot sulfur work on samples from Cahal Pech. We also thank Richard George, Brendan Culleton, and two anonymous reviewers who provided helpful feedback on earlier drafts and edits to this paper, which has made it a stronger contribution.

## Author Contributions

**Conceptualization:** Claire E. Ebert, Asta J. Rand, Kirsten Green-Mink.

**Data curation:** Claire E. Ebert, Kirsten Green-Mink, Julie A. Hoggarth, Jaime J. Awe, Willa R. Trask, Marie Danforth.

**Formal analysis:** Claire E. Ebert, Asta J. Rand, Kirsten Green-Mink.

**Funding acquisition:** Claire E. Ebert, Asta J. Rand, Julie A. Hoggarth, Jaime J. Awe, Douglas J. Kennett.

**Investigation:** Claire E. Ebert, Asta J. Rand, Jason Yaeger, M. Kathryn Brown.

**Methodology:** Claire E. Ebert, Asta J. Rand, Carolyn Freiwald.

**Project administration:** Claire E. Ebert.

**Resources:** Julie A. Hoggarth, Jaime J. Awe, Douglas J. Kennett.

**Supervision:** Douglas J. Kennett.

**Visualization:** Claire E. Ebert, Asta J. Rand, Carolyn Freiwald.

**Writing – original draft:** Claire E. Ebert, Asta J. Rand.

**Writing – review & editing:** Claire E. Ebert, Asta J. Rand, Kirsten Green-Mink, Julie A. Hoggarth, Carolyn Freiwald, Jaime J. Awe, Willa R. Trask, Jason Yaeger, M. Kathryn Brown, Christophe Helmke, Rafael A. Guerra, Marie Danforth, Douglas J. Kennett.

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
