## [Decision Letter · Decision Letter 0]

29 Mar 2021

PONE-D-21-05019

Applying Sulfur Isotopes to Paleodietary Reconstruction and Human Mobility from the Preclassic through Colonial Periods in the Eastern Maya Lowlands

PLOS ONE

Dear Dr. Ebert,

Thank you for submitting your manuscript to PLOS ONE. After careful consideration, we feel that it has merit but does not fully meet PLOS ONE’s publication criteria as it currently stands. Therefore, we invite you to submit a revised version of the manuscript that addresses the points raised during the review process.

Both reviewers agree on the quality and interest of the manuscript. They nevertheless raised important points that need to be addressed, such as the high variability in 13C and 34S abundances that could be expected from freshwater resources, the limitation of the lack of local baseline to elaborate on human diet composition or TEF values for 34S, and the terminology used when describing the isotopic results. I concur with reviewer 1 that 6‰ is an overestimated value of 15N TEF based on indirect and unconstrained isotopic spacing calculation. It is better to refer to studies which considered directly collagen to diet or collagen to collagen enrichment. I also agree with reviewer 2 that no predictive differences in 34S abundances can be stated for freshwater vs terrestrial resources in contrast with marine resources. As reviewer 2, I found some mistakes like 14N that should be 15N in the supporting information (e.g. paragraph 3.1)

I have also some concerns with the selection of the data considered as reliable. In the S1 Table, I noticed that you keep in Sulfur Group 2 samples with N:S lower than the 100 recommended by Nehlich and Richards (2009). As a matter of fact, these two collagen samples display a S content of 0.40 or higher. Nehlich and Richards (2009) recommend a S content range of 0.15 to 0.35% for mammal bones from archaeological contexts in addition to given C:S and N:S ranges. In Bocherens et al. 2011 (Palaeo, Paleaeo, Palaeo) the highest S content in modern mammals is found for reindeer with values as high as 0.33%. I would thus definitely not consider as secured the 34S abundances of collagen with a S content higher than 0.35%, as the authors state themselves in the supplementary data, which is the case for a number of samples of Sulfur group 2. Unless I am mistaken, it seems the authors have nevertheless not discarded these data from the final interpretation. This point needs to be addressed since all criteria of preservation have to be scrutinized and fulfilled before further elaborating on the meaning of the isotopic ratios.

I am puzzled by the collagen C content of 50% or higher which is over the expected value of even like-fresh collagen. I suspect you may have had some issues with the calibration of the elemental analysis. I would encourage you to check your measures comparing with the in-house standards with known elemental composition that were run in the same analytical sets.

We look forward to receiving your revised manuscript.

Kind regards,

Dorothée Drucker

Academic Editor

PLOS ONE

Journal Requirements:

2. In your manuscript, please provide additional information regarding the specimens used in your study. Ensure that you have reported specimen numbers and complete repository information, including museum name and geographic location.

For more information on PLOS ONE's requirements for paleontology and archaeology research, see https://journals.plos.org/plosone/s/submission-guidelines#loc-paleontology-and-archaeology-research

3. We note that Figures 1 and 2 in your submission contain map images which may be copyrighted.

a. You may seek permission from the original copyright holder of Figures 1 and 2 to publish the content specifically under the CC BY 4.0 license. 

Reviewers' comments:

Reviewer's Responses to Questions

**Comments to the Author**

1. Is the manuscript technically sound, and do the data support the conclusions?

Reviewer #1: Partly

Reviewer #2: Yes

2. Has the statistical analysis been performed appropriately and rigorously? 

Reviewer #1: Yes

Reviewer #2: Yes

3. Have the authors made all data underlying the findings in their manuscript fully available?

Reviewer #1: Yes

Reviewer #2: Yes

4. Is the manuscript presented in an intelligible fashion and written in standard English?

Reviewer #1: Yes

Reviewer #2: Yes

5. Review Comments to the Author

Reviewer #1: Please see page 2-8 of attached.

Meeting word limit. Meeting word limit.Meeting word limit. Meeting word limit. Meeting word limit. Meeting word limit. Meeting word limit. Meeting word limit.Meeting word limit.

Reviewer #2: This is a straightforward yet data-rich study that provides a set of valuable isotopic data from the Eastern Maya Lowland, including the first S measurements from the region. Other than improving our current understanding of past diet in the region, the data presented will also enrich the isotopic baselines of the Eastern Maya Lowlands, providing a solid base for future studies to build upon. I really enjoy reading this paper, the authors did a good job categorizing and analyzing a very complex set of data. This study should definitely be published after some minor revisions. Below, I have listed some comments that I hope the authors will find helpful. I also suggest the authors thoroughly proofread the writing to ensure correct scientific terminology is used, and make sure the correct studies are cited (I have spotted several mistakes below). For example, page 14, 2nd paragraph 9th line, it should be stable nitrogen isotope enrichment, not nitrogen enrichment. And page 16, last paragraph of section 5, should be “correlation between d15N and d34S values for subadults… “, instead of “d14N and d34S for sub-adults”. The word “values” need to be added throughout this section.

Minor and specific comments:

• Introduction: second line of second paragraph, “conclusive evidence” seems too strong, “compelling evidence”?

• Intro, 2nd paragraph: the second “Tykot et al. 2002” reference in the second paragraph seems misplaced, or perhaps the authors meant they are building upon the works done by Tykot et al?

• Last line of 3rd paragraph, other than Oelze et al. 2012 none of the study mentioned employed Sr isotope analysis, perhaps should rephrase the last sentence to “…other archaeological or isotope evidence”

• 4th paragraph 4th line: “…represents one of the largest…”

• Last paragraph in intro: need to clarify the correlation between δ15N and δ34S values are observed among subadults.

• Intro: Need to clarify that some of the data are from previously published reports, and also mention published Sr data will be used.

• Table 1 is excellent! It is really helpful for non-Mayan specialists like myself! Perhaps the authors can consider adding the number of samples analyzed from each period in this table?

• Section 2, 2nd paragraph, 4th line should be “the natural vegetations of the Maya lowlands…”

• Page 5 first line, I would argue that ¹³C values of freshwater fish as “highly variable” instead of “typically more negative”.

• Page 5 second paragraph, the reference of Cheung et al. 2019 is a mistake, that study did not look at S measurements.

• Result section: perhaps showing the figures with significant patterns (like through time, rather than by region?)

• Personally, I disagree that a low d34S values = “aquatic” sulfur isotope signature. d34S values in freshwater systems can vary greatly from region to region, some may have high d34S values.

• Page 5 3rd paragraph, the first sentence should be rewritten. I believe the authors are addressing the fractionation between consumer and source? In any case, 0.5 – 2.4‰ is a big range and not “slightly” different.

• Section 3.1, 1000 sq km is not a “small” geographic area.

• Figure 2 is a great map, but there isn’t much description in the text about how this model was produced. What parameters were involved? What statistical models and programs were used?

• Section 3.2, perhaps the authors can briefly explain why the following sites are selected (what are the site selection criteria?).

• Section 4.2 mentioned that %C and %N are used to evaluate bone collagen quality, but %C and %N are not showing in S1, please amend that.

• Table 3, is it possible to also add chronological info to this table?

• I recommend the authors show a figure comparing d13C values through time.

• As no faunal data is available, the discussion of trophic level seems inappropriate (section 5.1).

• Section 5.1, I personally don’t think it is a good idea to correct infant nitrogen isotope enrichment (Reynard et al’s paper certainly did not recommend that) to match the values of the adults.

• Section 5.2, I do not follow the discussion about “among terrestrial faunal species….”, please expand on this, or even include the faunal data point on the graph.

• Table 4, to me, the mean of group1 and 2 are so close that it is almost within analytical error, and group 4 only consists of 2 datapoints, so in reality, there should really be 2 groups?

• A figure showing correlation between the d13C and d34S values of the subadults could be interesting.

• Without any faunal data, I think it is inappropriate to discuss dietary compositions in too much specificity.

• Page 16 last line (to page 17 first line), be careful with citations here, Vika 2009 and Fornander 2013 only look at S measurements and did not use s isotope values to supplement results from strontium isotope analysis.

• Page 17, line 8, tropic should be trophic. However, as mentioned earlier, I don’t think with current data it is possible to discuss dietary compositions in such specificity.

• I think the comparison of d34S and 87Sr/86Sr values is a good idea! I wonder how the Sr measurements correlate with the geological map and isoscape model in figure 2?

• While the d34S values of the infants are interesting, I am not fully convinced of these “correlations”. The samples come from a vast temporal and geographical contexts, given that no faunal data is available to monitor regional patterns, and that the subadult sample size is very small, I recommend the authors be more subtle about their interpretations.

• Conclusion: I think the authors need to word the conclusion more carefully. As there is no faunal baseline data from the sites analysed, the authors need to tread very carefully and do not overinterpret their data. Freshwater fish especially, can have unpredictable/unexpected isotopic signature, therefore I don’t particular agree that d34S values can be used to say anything about the consumption of freshwater resources, at least not with current data. The second point is great, but kind of overlaps with the third point (did the outliers ate lots of non-local food or were they non-local themselves?). Perhaps the authors can rephrase this.

6. PLOS authors have the option to publish the peer review history of their article (what does this mean?). If published, this will include your full peer review and any attached files.

Reviewer #1: **Yes: **Eric Guiry

Reviewer #2: No

---

## [Author Response · Author response to Decision Letter 0]

4 Jun 2021

We have attached a document to our submissions with a detailed response to the reviewers' comments.

---

## [Decision Letter · Decision Letter 1]

29 Jun 2021

PONE-D-21-05019R1

Sulfur Isotopes as a Proxy for Human Diet and Mobility from the Preclassic through Colonial periods in the Eastern Maya Lowlands

PLOS ONE

Dear Dr. Ebert,

Thank you for submitting your manuscript to PLOS ONE. After careful consideration, we feel that it has merit but does not fully meet PLOS ONE’s publication criteria as it currently stands. Therefore, we invite you to submit a revised version of the manuscript that addresses the points raised during the review process.

I appreciate the efforts done by the authors to revise their manuscript. As far as preservation criteria are concerned, I have spotted one sample (MARC 2571 line 81 of excel table 1 S1) with a C:N ratio of 3.7 that should be excluded from further interpretation for the sake of consistency with the recommendation of De Niro (1985). About literature citation, I am not sure that the reference to O’Connell et al. (2012) is making sense to keep if you finally favour a more acknowledged TEF range of values unless you add “but see O’Connell et al., 2012” to mention possible debated on that point. Along the same lines, to my knowledge, Hedges and Reynard (2007) did not demonstrate a higher number of trophic levels in marine ecosystems. Pending the revision of the above points, the manuscript could be considered for publication.

We look forward to receiving your revised manuscript.

Kind regards,

Dorothée Drucker

Academic Editor

PLOS ONE

Journal Requirements:

Additional Editor Comments (if provided):

Reviewers' comments:

Reviewer's Responses to Questions

**Comments to the Author**

1. If the authors have adequately addressed your comments raised in a previous round of review and you feel that this manuscript is now acceptable for publication, you may indicate that here to bypass the “Comments to the Author” section, enter your conflict of interest statement in the “Confidential to Editor” section, and submit your "Accept" recommendation.

Reviewer #1: All comments have been addressed

2. Is the manuscript technically sound, and do the data support the conclusions?

Reviewer #1: Yes

3. Has the statistical analysis been performed appropriately and rigorously? 

Reviewer #1: Yes

4. Have the authors made all data underlying the findings in their manuscript fully available?

Reviewer #1: Yes

5. Is the manuscript presented in an intelligible fashion and written in standard English?

Reviewer #1: Yes

6. Review Comments to the Author

Reviewer #1: The revised manuscript appears to have addressed my comments. Well done. I have no further comments offer.

7. PLOS authors have the option to publish the peer review history of their article (what does this mean?). If published, this will include your full peer review and any attached files.

Reviewer #1: No

---

## [Author Response · Author response to Decision Letter 1]

29 Jun 2021

Please see attached "Response to Reviewers" document

---

## [Editor Report · Decision Letter 2]

8 Jul 2021

Sulfur Isotopes as a Proxy for Human Diet and Mobility from the Preclassic through Colonial periods in the Eastern Maya Lowlands

PONE-D-21-05019R2

Dear Dr. Ebert,

We’re pleased to inform you that your manuscript has been judged scientifically suitable for publication and will be formally accepted for publication once it meets all outstanding technical requirements.

A last remark is that the newly added reference Harriston and Harriston Am. Nat. 1993 needs to be corrected to Hairston and Hairston Am. Nat. 1993. Please proceed with this correction during the revision of the proofs.

Kind regards,

Dorothée Drucker

Academic Editor

PLOS ONE
---

## [Editor Report · Acceptance letter]

3 Aug 2021

PONE-D-21-05019R2 

Sulfur Isotopes as a Proxy for Human Diet and Mobility from the Preclassic through Colonial periods in the Eastern Maya Lowlands 

Dear Dr. Ebert:

I'm pleased to inform you that your manuscript has been deemed suitable for publication in PLOS ONE. Congratulations! Your manuscript is now with our production department. 

Kind regards, 

on behalf of

Dr. Dorothée Drucker 

Academic Editor

PLOS ONE